

# Scattering theory of higher order topological phases

R. Johanna Zijderveld[1*], Isidora Araya Day[1,2†] and Anton R. Akhmerov[1‡]

**1** Kavli Institute of Nanoscience, Delft University of Technology,
2600 GA Delft, The Netherlands
**2** QuTech, Delft University of Technology, Delft 2600 GA, The Netherlands

⋆ johanna@zijderveld.de , † iarayaday@gmail.com , ‡ scattering-hotis@antonakhmerov.org

## Abstract

The surface states of intrinsic higher order topological phases are protected by the spatial symmetries of a finite sample. This property makes the existing scattering theory of topological invariants inapplicable: the scattering geometry is either incompatible with the symmetry or does not probe the bulk topology. We resolve this obstacle by using a symmetric scattering geometry that probes transport from the inside to the outside of the sample. We demonstrate that the intrinsic higher order topology is captured by the flux dependence of the reflection matrix. Our finding follows from identifying the spectral flow of a flux line as a signature of higher order topology. We show how this scattering approach applies to several examples of higher order topological insulators and superconductors. Our theory provides an alternative approach for proving bulk–edge correspondence in intrinsic higher order topological phases, especially in the presence of disorder.

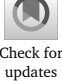

# 1  Introduction

The protection of surface states in higher order topological phases is more subtle than in topological insulators. Similar to strong topological insulators, higher order topological insulators (HOTIs) host surface states protected by local symmetries (time-reversal, particle-hole, or chiral), but they rely on additional spatial symmetries [1]. While in strong topological phases the existence of the edge state is guaranteed on the whole surface, in HOTIs any point of the surface may be gapped and the surface state may be freely moved around. In two dimensions, for example, a second order topological insulator with chiral symmetry hosts zero-dimensional corner states at zero energy [2]. If this system has reflection symmetries, these states may be removed by modifying the lattice termination—such phases are called extrinsic HOTIs [3–5]— shown in Fig. 1(a). An intrinsic HOTI, on the other hand, hosts corner states protected by the bulk–boundary correspondence. In two dimensions, for example, the combination of chiral and four-fold rotation guarantees that changing the surface of the sample in a symmetric way may not remove the corner states, see Fig. 1(b).

Despite the different nature of the surface states, important properties of topological insulators also apply to higher order phases. First, some higher order phases are stable in the presence of disorder, similar to strong topological phases [6–10]. This protection follows from the observation that removing a protected surface state requires coupling it with its symmetry partner, which, however, is located at the opposite edge of the sample. Therefore, because disorder acts locally, it does not remove the protected surface states. Secondly, several works used flux response and dislocation defects as a signature of higher order topology [11–18], generalizing the Laughlin argument. This is reminiscent of the spectral flow in strong topological phases. We thus identify two open questions:

- Can we characterize the topology of a HOTI in the presence of disorder?

- Is there a correspondence between surface states existing at boundaries of symmetric samples and flux response?

We adopt the scattering perspective to answer both questions. The scattering approach considers a transport setup and studies the reflection matrix from the surface of a finite sample. Previous works demonstrated that the reflection matrix at the Fermi level encodes the topological invariant of strong topological insulators [19, 20]. The topology of the reflection matrix changes simultaneously with the appearance of perfect transmission through a finite sample. Therefore, the scattering formalism proves that the changes of the scattering invariant—the invariant determined using the reflection matrix—are accompanied by delocalization transitions in disordered samples. Deep within the localized phase the reflection matrix is unitary and it contains information about the low energy spectrum of the boundary. In a topological system, this probes the anomalous Hamiltonian of the protected surface states. Furthermore,

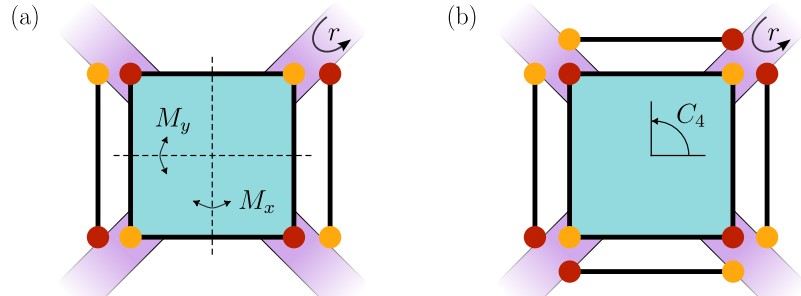

Figure 1: Extrinsic and intrinsic higher order phases in the BBH model. The leads (purple) are attached to the corners of the sample to probe the corner states in the different sublattices (red and orange) via the reflection matrix $r$. (a) An extrinsic phase has states that may be removed by changing the lattice termination (vertical chains) (b) An intrinsic phase has states that are protected by the bulk topology and cannot be removed by changing the lattice termination.

because this approach starts from a real space description of the system, scattering invariants allow to study topological phases in amorphous [21] and quasi-crystalline materials. Finally, determining the reflection matrix is more computationally efficient than obtaining the full spectrum [22], making this approach competitive.

A direct application of the scattering formalism to higher order topological insulators is to consider a finite sample and to attach leads where the zero modes may be located [3, 23, 24]. The scattering invariant computed this way may change whenever transmission between disconnected leads appears, which may happen both when the bulk gap closes and when the surface gap closes. The sensitivity to the surface gap glosing makes this approach suitable to probe the topology of the extrinsic HOTIs, but makes its applicability to intrinsic HOTIs unclear. Due to the reliance on the protection by the surface gap, we call the established approach an *extrinsic scattering invariant*, and we demonstrate that it fails when applied to intrinsic HOTIs using the Benalcázar–Bernevig–Hughes (BBH) model [25] as an example.

We resolve the limitation of the scattering theory of topological invariants and develop its extension to intrinsic HOTIs by defining the *intrinsic scattering invariants*. We demonstrate that our approach correctly classifies the topological phases in a disordered BBH model. We then extend the approach to other second order HOTIs, and answer positively to the question of the existence of a relation between higher order topology and flux response.

## 2 Why the extrinsic scattering invariant is not enough

The most direct approach to applying a scattering invariant to a HOTI is to use the appearance of the protected zero modes on the edge. This requires probing a reflection matrix of a finite symmetric sample with leads attached to its edges in a symmetric way. Symmetry class–dependent functions of the reflection matrix at each lead count the number of protected zero modes at the edge, and their parity was used as a scattering probe for HOTIs [3]. This extrinsic scattering invariant, and similar real space HOTI invariants [26, 27], rely on knowing the position of protected edge states, and therefore they are limited in usefulness for characterizing an unknown system.

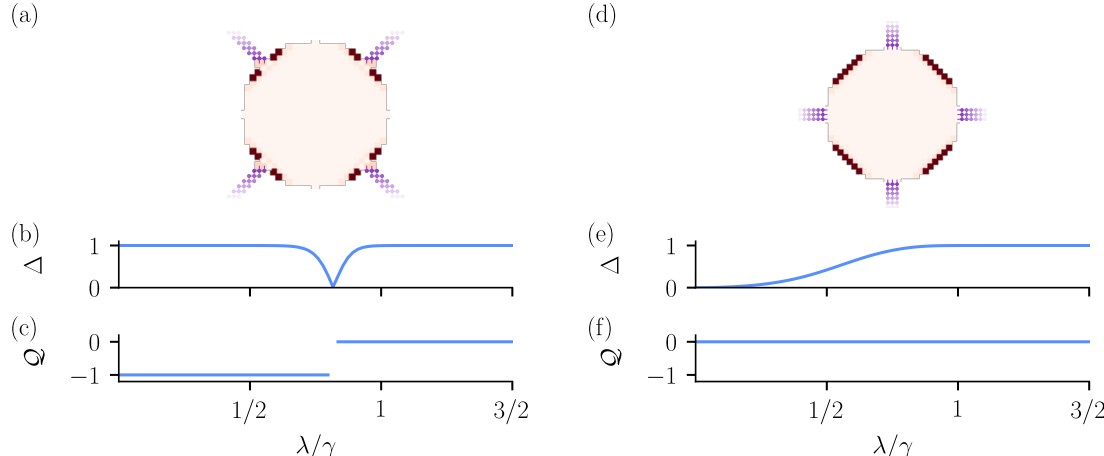

Figure 2: Scattering geometries that probe the topology of a two-dimensional intrinsic higher order topological insulator using an extrinsic scattering invariant. Panels (a) and (d) show the local charge density of the zero modes in the scattering region with leads (purple) attached in two different configurations. The gap of the reflection matrix $\Delta = |\det r|$ in (b/e) and the signature of $r$ in (c/f)—the invariant $\mathcal{Q}$—are sensitive to the location of the leads.

To illustrate the limitations of the extrinsic scattering invariant, we consider an example two-dimensional intrinsic HOTI with four-fold rotation anticommuting with sublattice symmetry, known as the BBH model [25]. We construct a circular sample, attach four leads[1] to its edges, and compute the reflection matrix $r$ of a single lead as a function of model parameters. Applying the sublattice symmetry constraint, we transform $r$ to the basis where $r = r^\dagger$. In this basis, the topological invariant $\mathcal{Q}$ is the signature of the reflection matrix [19]—the difference between the number of positive and negative eigenvalues of $r$.

We characterize the topological transition by additionally computing the gap of the reflection matrix: $\Delta = |\det r|$. This gap is a useful probe of scattering topological invariants because it must vanish simultaneously with the topological invariant changing. Figure 2 shows the signature $\mathcal{Q}$ and the reflection gap $\Delta$ as a function of $\lambda/\gamma$, with $\lambda$ the intra-cell hopping and $\gamma$ the inter-cell hopping of the BBH model. The value $\lambda/\gamma = 1$ corresponds to the topological phase transition of the BBH model [25], which is shifted in a finite sample due to finite-size effects. We observe that the topological invariant strongly depends on the sample geometry. When the leads are attached at the points where the zero modes are located, the invariant changes at the phase transition as expected [Fig. 2(a-c)]. On the other hand, if the leads are attached exactly between the zero modes, as shown in Fig. 2(e-f), the invariant stays constant across the phase transition.

This failure of the extrinsic scattering invariant to probe the topology of the intrinsic HOTI is a consequence of the invariant being sensitive to the precise location of the zero modes as well as the possibility for the invariant to change without the bulk gap closing. The same limitation likely applies to several other real space probes of HOTI phases, such as localizer invariant [28], multipole winding number [26], mode–shell correspondence [29], and Bott index [30].

---

[1]Throughout the manuscript we use ideal leads, as defined in Ref. [20].

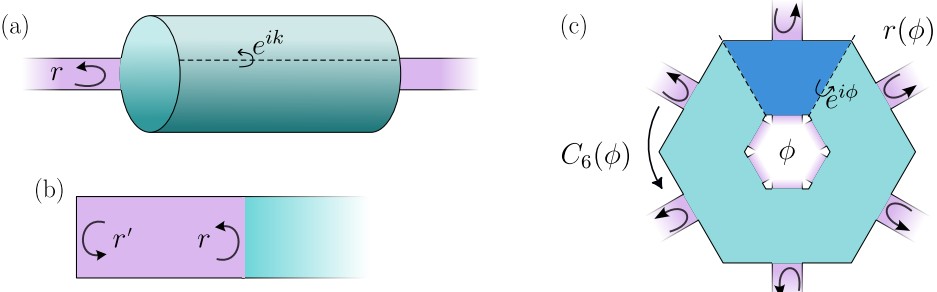

Figure 3: Scattering geometries that probe the existence of zero-energy surface states using the bulk symmetries. The reflection matrix $r$ encodes the amplitudes of the wavefunctions that are reflected back into the lead (purple) from the scattering region (turquoise). (a) Scattering setup used to probe topological phases protected by local symmetries. A two-dimensional bulk Hamiltonian defines a one-dimensional reflection matrix by attaching leads along one direction and applying twisted boundary conditions in the other. The reflection matrices $r$ and $r'$ capture the appearance of zero-energy bound states (b). (c) Scattering setup used to probe topological phases protected by spatial symmetries. A two-dimensional six-fold symmetric bulk Hamiltonian defines a one-dimensional reflection matrix by attaching symmetric leads and introducing a flux through the center of the scattering region. The blue area shows the matrix elements of the symmetry operator that gain a phase factor due to the flux, defining $C_6(\phi)$. The scattering setup probes the reflection from the outer lead, which is composed of six disjoint parts related to each other by the rotation symmetry.

## 3 Intrinsic scattering invariant construction

The scattering matrix $S(E)$ relates the incoming and outgoing wavefunctions between the leads at an energy $E$, and its elements are given by the scattering equation:

$$(H - E)\left(\Psi^{\text{in}}\alpha^{\text{in}} + \Psi^{\text{out}}S(E)\alpha^{\text{in}} + \Psi^{\text{localized}}\alpha^{\text{in}}\right) = 0, \tag{1}$$

where $H$ is the Hamiltonian of the scattering region and the leads, and $\Psi^{\text{in}}$, $\Psi^{\text{out}}$, and $\Psi^{\text{localized}}$ are the wavefunctions of the incoming, outgoing, and localized states, respectively. These wavefunctions are matrices with columns corresponding to different modes, see App. A for a detailed description. To treat topology due to antisymmetries that map $E$ to $-E$ on the same footing, we set the energy $E$ to zero. The incoming and outgoing wavefunctions occupy the same lead sites and orbitals, while the localized wavefunctions are only defined in the scattering region. The total number of modes is the number of columns of the wavefunction matrices $N_{\text{modes}} = N_{\text{sites}} \times N_{\text{orbitals}}$, where $N_{\text{sites}}$ is the number of sites to which the leads are attached, and $N_{\text{orbitals}}$ is the number of orbitals per site in the lead. The scattering equations define a valid solution for any combination of amplitudes $\alpha^{\text{in}}$ of the incoming modes. Because the scattering matrix in Eq. (1) specifies a relation between the amplitudes of the incoming and outgoing wavefunctions, the scattering matrix is a linear map rather than an operator. Separating the leads into two groups imposes a block structure on the scattering matrix:

$$S = \begin{pmatrix} r & t' \\ t & r' \end{pmatrix}, \tag{2}$$

where $r$, $r'$ are reflection matrices, and $t$, $t'$ are the transmission matrices.

The scattering theory of topological invariants [20] relates the topology of a $d$-dimensional bulk Hamiltonian $H_{\text{bulk}}(\boldsymbol{k}_d)$, with $\boldsymbol{k}_d$ the wave vector, to the topology of a $d-1$-dimensional effective Hamiltonian $H_{\text{eff}}(\boldsymbol{k}_{d-1})$. The dimensional reduction procedure to obtain this effective Hamiltonian is as follows:

1. Construct a large finite sample with the Hamiltonian of the scattering region equal to $H_{\text{bulk}}$.

2. Attach two leads along one of the dimensions and apply twisted periodic boundary conditions along all other dimensions, as shown in Fig. 3(a). The phases along each dimension define the new wave vector $\boldsymbol{k}_{d-1}$.

3. Compute the reflection matrix $r(\boldsymbol{k}_{d-1})$ of one of these leads.

4. Choose a proper symmetry representation so that the symmetry constraints on $r(\boldsymbol{k}_{d-1})$ simplify.

5. Define $H_{\text{eff}}(\boldsymbol{k}_{d-1})$ using:
$$H_{\text{eff}} = \begin{cases} r = r^\dagger, & \text{if } H_{\text{bulk}} \text{ has chiral symmetry,} \\ \begin{pmatrix} 0 & r \\ r^\dagger & 0 \end{pmatrix}, & \text{otherwise.} \end{cases} \tag{3}$$

This effective Hamiltonian has zero eigenvalues simultaneously with $r(\boldsymbol{k}_{d-1})$ and a symmetry class different from the symmetry of $H_{\text{bulk}}$. The zero eigenvalues of $r(\boldsymbol{k}_{d-1})$ correspond to quantized transmission eigenvalues from one lead to the other, which only happens if the gap of $H_{\text{bulk}}$ closes. Because of this relation, the topological invariants of $H_{\text{eff}}$ and $H_{\text{bulk}}$ are equal. An alternative interpretation of the scattering invariant is to consider a boundary between the topological bulk and a trivial region. In this case, the boundary Green's function defines the topology of the bulk [31]. In the scattering description this boundary corresponds to interrupting the lead with a reflection matrix $r'$ of the trivial region, as shown in Fig. 3(b). The zero energy modes at an interface of an infinite system appear whenever $r r'$ has an eigenvalue equal to 1, and therefore the protected zero energy solutions at the interface are encoded in $r$.

While the procedure above may be applied to intrinsic HOTIs, the resulting scattering geometry either has leads swapped by the symmetry of the HOTI, or is incompatible with the global spatial symmetry protecting the HOTI phase. This incompatibility makes the topology of $H_{\text{eff}}$ different from the higher order topology of $H_{\text{bulk}}$. Instead, in order to apply the scattering approach to an intrinsic HOTI, the scattering geometry must be compatible with the global spatial symmetry protecting the HOTI phase, and this symmetry must map the leads onto themselves. At the same time, the two leads must be separated by the bulk rather than belonging to the same surface, such that no topological boundary modes propagate between them. This is required in order for zeros of the reflection matrix to correspond to the bulk gap closing and therefore to probe only bulk topological transitions. Otherwise, zeros of the reflection matrix could appear due to the closing of the surface gap and therefore not probe the bulk topology, as illustrated in Sec. 2. To satisfy the requirements of the scattering theory we propose to use a finite geometry with a hole: an annulus in two dimensions or a cylinder in three dimensions, and attach one lead to the inside and one lead to the outside of the sample. Figure 3(c) shows such a scattering geometry with both the inner and outer leads consisting of several disjoint parts that are related among themselves by the rotating symmetry. Whether the leads are attached to the entire circumference or only a part of it changes the magnitude of the finite size effects, but keeps the topological properties of the scattering geometry the same.

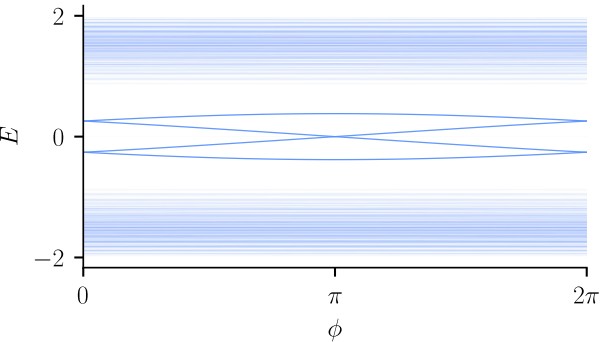

Figure 4: Energy spectrum of a topological BBH model in an annulus geometry as a function of the threaded flux $\phi$. The transparency of the bands is proportional to the weight of the wavefunctions on the inner ring of the annulus, such that the four states closest to the Fermi level are localized on the inner ring, while the states at the outer ring are not shown. Sublattice symmetry makes the positive and negative energies symmetric, pinning the states crossing to zero energy, but not at any specific value of $\phi$—the crossing at $\phi = \pi$ is fine-tuned. The spectral flow of the zero modes is only present in the topological phase. The annulus used for the computation has an inner radius $r_{\text{inner}} = 1$ and an outer radius $r_{\text{outer}} = 10$, $\lambda/\gamma = 0.4$.

Similar to the dimensional reduction procedure, we apply twisted boundary conditions in the direction parallel to the cylinder axis, which introduces a wave vector $k_{d-2}$. Without introducing additional parameters, this would result in a $d-2$-dimensional reflection matrix, which loses information about the angular direction along the cylinder and is unlikely to be sufficient to determine the HOTI topology. To preserve the information about the angular direction, we therefore introduce a flux $\phi$ through the center of the cylinder, which acts as an additional momentum parameter. The resulting procedure maps a $d$-dimensional bulk onto a $d-1$-dimensional $H_{\text{eff}}(\phi, \boldsymbol{k}_{d-2})$, similar to the dimensional reduction of the strong topological insulators. Contracting the inner radius of the cylinder to a point generally gaps out the surface modes due to finite size effects. Therefore, the flux is also necessary to cancel this splitting, as shown in Fig. 4. This cancellation of the finite size splitting was reported in Ref. [32], in a specific symmetry class. By confirming that the scattering invariants work in the annulus geometry, we show that the flux response is a general feature of the higher order topology.

We derive the constraints on the scattering matrix $S$ by considering all the symmetries of the Hamiltonian $H$ and separately applying each of them to the scattering matrix. Applying an operator $\mathcal{O}$ which may be unitary or antiunitary to the scattering equation gives us the transformed scattering equation:

$$\tilde{H}(R_{\mathcal{O}}(\phi, \boldsymbol{k}_{d-2}))\left(\mathcal{O}\Psi^{\text{in}}\alpha^{\text{in}} + \mathcal{O}\Psi^{\text{out}}S(\phi, \boldsymbol{k}_{d-2})\alpha^{\text{in}} + \mathcal{O}\Psi^{\text{localized}}\alpha^{\text{in}}\right) = 0, \qquad (4)$$

where $\tilde{H}(R_{\mathcal{O}}(\phi, \boldsymbol{k}_{d-2})) = \mathcal{O}H(\phi, \boldsymbol{k}_{d-2})\mathcal{O}^{-1}$ is the transformed Hamiltonian, and $R_{\mathcal{O}}$ is the action of the operator in parameter space, which in general depends on flux and momenta. To obtain the transformed scattering matrix $\tilde{S}(R_{\mathcal{O}}(\phi, \boldsymbol{k}_{d-2}))$, we act with the operator on $\mathcal{O}$ on the wavefunctions and coefficients of the wavefunctions, and equate the coefficients of the wavefunctions in Eq. (1) and Eq. (4) (see App. B for a detailed derivation). The action of an operator on the lead wavefunctions depends on whether this operator is unitary or antiunitary, and whether it is a symmetry or an antisymmetry of the Hamiltonian. We denote the unitary symmetries $\mathcal{U}$, unitary antisymmetries $\mathcal{C}$ (chiral symmetry), antiunitary symmetries $\mathcal{T}$ (time-reversal symmetry), and antiunitary antisymmetries $\mathcal{P}$ (particle-hole symmetry). The action

of these 4 types of operators on the lead modes is [20]:[2]

$$\mathcal{U}\Psi^{\text{in}} = \Psi^{\text{in}}V_{\mathcal{U}}(\phi, \boldsymbol{k}_{d-2}), \qquad \mathcal{U}\Psi^{\text{out}} = \Psi^{\text{out}}Q_{\mathcal{U}}(\phi, \boldsymbol{k}_{d-2}), \tag{5a}$$

$$\mathcal{P}\Psi^{\text{in}} = \Psi^{\text{in}}V_{\mathcal{P}}(\phi, \boldsymbol{k}_{d-2}), \qquad \mathcal{P}\Psi^{\text{out}} = \Psi^{\text{out}}Q_{\mathcal{P}}(\phi, \boldsymbol{k}_{d-2}), \tag{5b}$$

$$\mathcal{T}\Psi^{\text{in}} = \Psi^{\text{out}}V_{\mathcal{T}}(\phi, \boldsymbol{k}_{d-2}), \qquad \mathcal{T}\Psi^{\text{out}} = \Psi^{\text{in}}Q_{\mathcal{T}}(\phi, \boldsymbol{k}_{d-2}), \tag{5c}$$

$$\mathcal{C}\Psi^{\text{in}} = \Psi^{\text{out}}V_{\mathcal{C}}(\phi, \boldsymbol{k}_{d-2}), \qquad \mathcal{C}\Psi^{\text{out}} = \Psi^{\text{in}}Q_{\mathcal{C}}(\phi, \boldsymbol{k}_{d-2}), \tag{5d}$$

where $V_{\mathcal{O}}(\phi, \boldsymbol{k}_{d-2})$ and $Q_{\mathcal{O}}(\phi, \boldsymbol{k}_{d-2})$ are unitary matrices, which we compute by projecting the operator $\mathcal{O}$ onto the lead wavefunctions. By combining the transformed scattering equation Eq. (4) with the transformed wavefunctions in Eqs. (5), we obtain the transformed reflection matrix $\tilde{r}$:

$$\tilde{r}(R_{\mathcal{U}}(\phi, \boldsymbol{k}_{d-2})) = Q_{\mathcal{U}}(\phi, \boldsymbol{k}_{d-2})r(\phi, \boldsymbol{k}_{d-2})V_{\mathcal{U}}^{\dagger}(\phi, \boldsymbol{k}_{d-2}), \tag{6a}$$

$$\tilde{r}(-\phi, -\boldsymbol{k}_{d-2}) = Q_{\mathcal{P}}(\phi, \boldsymbol{k}_{d-2})r^{*}(\phi, \boldsymbol{k}_{d-2})V_{\mathcal{P}}^{\dagger}(\phi, \boldsymbol{k}_{d-2}), \tag{6b}$$

$$\tilde{r}(-\phi, -\boldsymbol{k}_{d-2}) = V_{\mathcal{T}}(\phi, \boldsymbol{k}_{d-2})r^{T}(\phi, \boldsymbol{k}_{d-2})Q_{\mathcal{T}}^{\dagger}(\phi, \boldsymbol{k}_{d-2}), \tag{6c}$$

$$\tilde{r}(\phi, \boldsymbol{k}_{d-2}) = V_{\mathcal{C}}(\phi, \boldsymbol{k}_{d-2})r^{\dagger}(\phi, \boldsymbol{k}_{d-2})Q_{\mathcal{C}}^{\dagger}(\phi, \boldsymbol{k}_{d-2}). \tag{6d}$$

If $\mathcal{O}$ is a symmetry, then $\tilde{r} = r$ and Eqs. (6) provide symmetry constraints on $r$.

Due to $r(\phi, \boldsymbol{k}_{d-2})$ being a linear map (as a submatrix of $S$), the symmetry operations (6) apply to $r(\phi, \boldsymbol{k}_{d-2})$ differently than to $H_{\text{bulk}}$. Additionally, due to Eq. (3), whenever $H_{\text{bulk}}$ has chiral symmetry, $H_{\text{eff}}$ does not, and vice versa. This results in the Altland–Zirnbauer symmetry class of $H_{\text{bulk}}$ being related to that of $H_{\text{eff}}$ according to the Bott periodicity [20]. For example, in the symmetry class D ($\mathcal{P}^2 = 1$), particle-hole symmetry constraint is equivalent to $H_{\text{bulk}}(\mathbf{k}) = -H_{\text{bulk}}^{*}(-\mathbf{k})$ up to a basis choice. The reflection matrix symmetry constraint (6b) is then equivalent to $r(-\phi, \boldsymbol{k}_{d-2}) = r^{*}(-\phi, -\mathbf{k}_{d-2})$. After applying (3), the effective Hamiltonian belongs to the symmetry class BDI ($\mathcal{T}^2 = \mathcal{P}^2 = 1$).

The first step in determining how the spatial unitary symmetries apply to the effective Hamiltonian is to identify the transformation $R_{\mathcal{U}}(\phi, \boldsymbol{k}_{d-2})$ under the symmetry operator $\mathcal{U}$. The scattering geometry of Fig. 3(c) respects all symmetries that leave the rotation axis invariant. The momenta $\boldsymbol{k}_{d-2}$ transform as vectors while the flux $\phi$ transforms as a pseudovector pointing along the rotation axis. Translations and rotations both keep $(\phi, \boldsymbol{k}_{d-2})$ invariant. Inversion symmetry changes the sign of all vector components, while keeping pseudovectors the same:

$$R_{\mathcal{I}}(\phi, \boldsymbol{k}_{d-2}) = (\phi, -\boldsymbol{k}_{d-2}). \tag{7}$$

Finally, a reflection symmetry $M$ results in

$$R_M(\phi, \boldsymbol{k}_{d-2}) = (\mp\phi, \pm\boldsymbol{k}_{d-2}), \tag{8a}$$

with the sign depending on whether $M$ commutes or anticommutes with rotation around the axis. The next step requires constructing a symmetry operator in presence of flux $\phi$ inserted at the rotation axis. We choose a gauge where the flux $\phi$ enters $H_{\text{bulk}}$ as a phase $\exp i\phi$ for all hoppings across a branch cut, as shown in Fig. 3(c). Spatial rotations rotate this branch cut by $2\pi/n$, with $n$ the number of rotations needed to do one full rotation, and inversion rotates it by $\pi$. Therefore, to keep the Hamiltonian invariant, wavefunctions in the scattering region between the initial and final position of the branch cut acquire a phase $\exp i\phi$, as shown in Fig. 3(c). We therefore find that the operator $C_n$ of the rotation by $2\pi/n$ exponentiates to a flux dependent value: $C_n^n(\phi) = C_n^n(0)\exp i\phi$, and accordingly the representations

---

[2]Note that we choose a different convention for $V$ and $Q$ than in Ref. [20]. Swapping $V \leftrightarrow V^T$ and $Q \leftrightarrow Q^T$ would make our convention equivalent to Ref. [20].

of the rotation symmetry $V_{C_n}, Q_{C_n}$ then satisfy $V_{C_n}^n = C_n^n(0) \exp i\phi$ and $Q_{C_n}^n = C_n^n(0) \exp i\phi$. Similarly, inversion rotates the branch cut by $\pi$ and squares to $\mathcal{I}^2(\phi) = \mathcal{I}^2(0) \exp i\phi$, so that $V_{\mathcal{I}}^2 = \mathcal{I}^2(0) \exp i\phi$ and $Q_{\mathcal{I}}^2 = \mathcal{I}^2(0) \exp i\phi$. The periodic dependence $r(\phi) = r(\phi + 2\pi)$ invites interpreting $\phi$ as a momentum of $H_{\text{eff}}$ with the Brillouin zone spanned by $\boldsymbol{k}_{d-2}$ and $\phi$. Within this parallel, a $C_n$ rotation symmetry of $H_{\text{bulk}}$ which increments the polar angle by $2\pi/n$, results in a translation symmetry by $1/n$ of a unit cell of $H_{\text{eff}}$. Likewise, inversion symmetry results in a combination of reflection with respect to the plane perpendicular to the rotation axis and translation by half a unit cell in $H_{\text{eff}}$. In other words, inversion of the original scattering geometry acts as a glide symmetry on $H_{\text{eff}}$.

Rotations, inversions, and reflections are unitary symmetries, and therefore the symmetry constraints on the reflection matrix $r(\phi, \boldsymbol{k}_{d-2})$ are given by Eq. (6a). More general spatial symmetries, like the $C_4\mathcal{T}$ symmetry that protects the HOTI phase in three dimensions [1], are a combination of a unitary operation with an antiunitary transformation or an antisymmetry. We derive the symmetry constraints on the reflection matrix $r$ by applying the individual operators in the symmetry sequentially, and then combining the results. For example, the $C_4\mathcal{T}$ symmetry is composed of the four-fold rotation $C_4$ and time-reversal $\mathcal{T}$ operators, under which the reflection matrix $r$ transforms according to Eqs. (6a) and (6c) respectively. Therefore, the transformation of the reflection matrix under $C_4\mathcal{T}$ is given by the product of the transformations under $C_4$ and $\mathcal{T}$:

$$\tilde{r}(-\phi, -\boldsymbol{k}_{d-2}) = V_{C_4\mathcal{T}}(\phi, \boldsymbol{k}_{d-2}) r^T(\phi, \boldsymbol{k}_{d-2}) Q_{C_4\mathcal{T}}^\dagger(\phi, \boldsymbol{k}_{d-2}), \tag{9}$$

where we define $V_{C_4\mathcal{T}} = V_{C_4}(\phi, \boldsymbol{k}_{d-2}) V_{\mathcal{T}}(\phi, \boldsymbol{k}_{d-2})$ and $Q_{C_4\mathcal{T}} = Q_{\mathcal{T}}(\phi, \boldsymbol{k}_{d-2}) Q_{C_4}(\phi, \boldsymbol{k}_{d-2})$. Once again, if $C_4\mathcal{T}$ is a symmetry, then $\tilde{r} = r$ and the symmetry constraints on $r$ are given by the above equation.

In practice, the first step in determining the topological invariant requires finding a basis for the incoming and outgoing wavefunctions where the symmetry constraints on the reflection matrix are minimal. Choosing an appropriate basis simplifies the time-reversal symmetry constraint to either symmetry or antisymmetry of $r(\phi, \boldsymbol{k}_{d-2})$, particle-hole symmetry becomes a reality constraint, and chiral symmetry becomes a Hermiticity constraint, as shown in App. B. In the presence of multiple symmetries, we choose the basis where the chiral symmetry gives a Hermiticity constraint if it is present, according to Eq. (3), and the other symmetries have a minimal compatible form, following Ref. [20]. After performing this transformation, we rely on the established theory of Hamiltonian topological invariants to determine the topological invariant of $H_{\text{eff}}$.

## 4 Applications of the intrinsic scattering invariant

To confirm the universality of the procedure presented in the previous section, we demonstrate how it applies to several important examples of HOTIs. These examples illustrate the ways in which the spatial symmetries of the HOTI phase translate into a symmetry group of the dimensionally reduced Hamiltonian. While we do not aim to provide a complete catalog of scattering invariants for HOTIs, this section presents a strong argument in favor of the universality of the procedure.

### 4.1 Intrinsic scattering invariant of BBH model

To construct the intrinsic scattering invariant of the BBH model, we first consider the reflection matrix of an annulus-shaped lattice with a flux $\phi$ and inner and outer leads, as

shown in Fig. 5(a). In the topological phase, both surfaces of the annulus host four zero-dimensional protected zero modes. Four-fold rotation and sublattice symmetries given by Eqs. (6a) and (6d) constrain the reflection matrix $r(\phi)$ as

$$r(\phi) = Q_{C_4}(\phi)\, r(\phi) V_{C_4}^\dagger(\phi) = V_\mathcal{C}\, r^\dagger(\phi) Q_\mathcal{C}^\dagger\,, \tag{10}$$

where $V_{C_4}^4(\phi) = Q_{C_4}^4(\phi) = -\exp i\phi$ acquire a phase factor due to the flux $\phi$, as explained in Sec. 3. Additionally, from $\mathcal{C}^2 = 1$ we obtain that $V_\mathcal{C}^\dagger = Q_\mathcal{C}$. Combining the anticommutation $C_4\mathcal{C} = -\mathcal{C}C_4$ with Eqs. (5a,5d) additionally yields

$$V_\mathcal{C} V_{C_4}(\phi) = -Q_{C_4}(\phi) V_\mathcal{C}\,. \tag{11}$$

With all the constraints, we are ready to determine the intrinsic scattering invariant of the BBH model.

As the first step, we simplify the problem by observing that $C_2 \equiv C_4^2$ commutes both with $\mathcal{C}$ and $\mathcal{C}_4$. Because $C_2^2(\phi) = -\exp(i\phi)$, interpreting $\phi$ as a momentum parameter of the effective Hamiltonian means that the problem is invariant under translation by half a unit cell in the corresponding direction. Therefore, we unfold the Brillouin zone and reduce the unit cell to a primitive one to eliminate redundant degrees of freedom. Because the eigenvalues of $C_2$ are $\pm i \exp(i\phi/2)$, they are periodic over $4\pi$ and the two subspaces of $C_2$—and of any operator that commutes with it—swap over a $2\pi$ interval. Therefore, we project $r$ and the remaining symmetries onto the subspace of $C_2$ with the eigenvalue $i \exp i\phi/2$, and consider their dependence on $\phi \in [0, 4\pi)$. The range of $\phi \in [2\pi, 4\pi)$ of the $i\exp i\phi/2$ eigenspace is equivalent to the range $\phi \in [0, 2\pi)$ of the $-i \exp i\phi/2$ eigenspace, so choosing one subspace over twice the interval provides an equivalent representation. To avoid dealing with $4\pi$ periodicity, we redefine $\phi := \phi/2$, which yields $r(\phi)$ that satisfies symmetry constraints equivalent to those in Eq. (10), but with $V_{C_4}^2 = Q_{C_4}^2 = i\exp i\phi$ instead. We call this procedure factoring out a symmetry, because $r(\phi)$ is no longer constrained by $C_2$. We describe how to factor out a symmetry and transform the remaining symmetries to the new basis in App. C.

We proceed simplifying the symmetry constrains by transforming to the eigenbasis of the remaining symmetry operators. We redefine $r(\phi) := V_\mathcal{C}^\dagger r(\phi)$ and $V_{C_4}(\phi) = -Q_{C_4}(\phi) := V_\mathcal{C} V_{C_4}(\phi)$, and simplify the symmetry constraints to

$$r(\phi) = -V_{C_4}(\phi) r(\phi) V_{C_4}^\dagger(\phi) = r^\dagger(\phi)\,. \tag{12}$$

The first equality corresponds to an anticommutation relation, making $r(\phi)$ block-offdiagonal in the basis of $V_{C_4}$. The second equality establishes that $r(\phi)$ is also Hermitian in this basis. We then transform $r(\phi)$ to the eigenbasis of $V_{C_4}$, and obtain the effective Hamiltonian

$$H_{\mathrm{eff}}(\phi) = \begin{pmatrix} 0 & h(\phi) \\ h^\dagger(\phi) & 0 \end{pmatrix}, \qquad h(\phi + 2\pi) = h^\dagger(\phi), \tag{13}$$

where the blocks of $H_{\mathrm{eff}}(\phi)$ are in the basis of the different eigensubspaces of $V_{C_4}$. The eigensubspaces of $V_{C_4}$ swap under $\phi \to \phi + 2\pi$, and therefore the effective Hamiltonian in Equation (13) is $4\pi$-periodic in $\phi$. Contrary to the dimensional reduction procedure used for strong topological insulators, this Hamiltonian has a sublattice-like symmetry—more specifically, a glide sublattice symmetry—just like its parent BBH model. The final step is to apply the topological invariant of $H_{\mathrm{eff}}(\phi)$, however, to the best of our knowledge, the topology of one-dimensional systems combining fractional translations and sublattice symmetries has not been studied before. Therefore, we proceed to construct the invariant, although in most cases one may apply the already established invariants to the effective Hamiltonian.

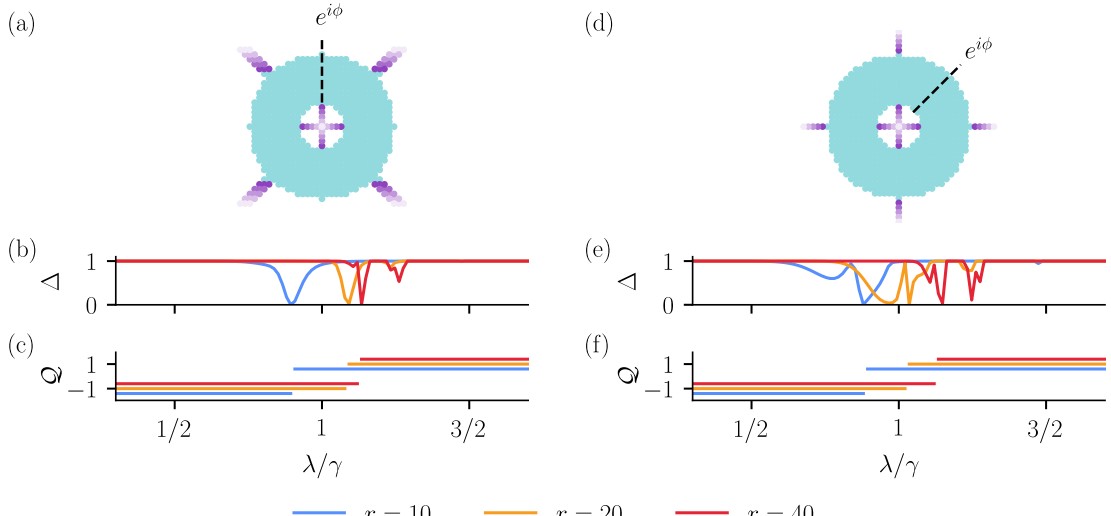

Figure 5: Two scattering geometries of the BBH model and their intrinsic invariant in the presence of globally symmetric disorder. (a/d) Annulus geometry with a flux $\phi$ threaded through the center and leads (purple) attached to the inner and outer rings. In (a) the leads are attached where the zero modes are located and in (d) the leads are in between the zero modes. The gap of the reflection matrix $\Delta = \min|\det r(\phi)|$ in (b/e) closes simultaneously with the sign change of the invariant $\mathcal{Q}$ in (c/f). The results are shown for different outer ring sizes $r = 10, 20, 40$, and in (c/f) the values of $\mathcal{Q}$ are minimally shifted for clarity.

Because $H_{\text{eff}}(\phi)$ is gapped as long as $h(\phi)$ has no zero eigenvalues, its topological invariant must be a function of $\det h(\phi)$. Because $\det h(\phi + 2\pi) = \det h^*(\phi)$, $\det h(\phi)$ crosses the real axis an odd number of times in the interval $\phi \in [0, 2\pi)$. The $\mathbb{Z}_2$ invariant of this system is the parity of the number of the crossings by $\det h(\phi)$ of the positive real axis. The crossings of the real axis appear and disappear only in pairs, hence the parity of the number of crossings changes only if $\det h(\phi) = 0$ for some $\phi$. Therefore we identify the scattering invariant of the BBH model to be:

$$\mathcal{Q} = \text{sign}\left(\det h(-\pi) \exp\left[\frac{1}{2}\int_{-\pi}^{\pi} d \log \det h(\phi)\right]\right) \in \mathbb{Z}_2. \tag{14}$$

Because to compute $h(\phi)$ we project $r(\phi)$ onto the subspaces of $C_2$ as described in App. C, we require special care to ensure that $H_{\text{eff}}(\phi)$ depends continuously on $\phi$. We address this need by computing an eigenvalue decomposition of $V_{C_2}(\phi)$ and $Q_{C_2}(\phi)$ that varies smoothly with $\phi$, as described in the App. D. At this point, we have constructed a scattering invariant that uses all of the spatial symmetries of the BBH model, an important difference from extrinsic scattering invariants. Additionally, the invariant requires knowledge of the reflection matrix at all values of $\phi \in [0, 2\pi)$, in agreement with the spectral flow in Fig. 4, where the zero-energy crossings of the BBH model are not pinned to any specific value of $\phi$.

We construct the model and compute the scattering matrix using Kwant [33]. To demonstrate that the scattering invariant applies to disordered systems, we add a $C_4$-symmetric disorder to the BBH model. Specifically, we choose the intra-cell hopping equal to $\lambda[1 + \delta(x, y)]$, where $\delta(x, y) = \delta(y, -x)$ is symmetric under $C_4$ rotations, and each $\delta(x, y)$ not related by symmetry is an independent normally distributed random variable with zero mean and standard deviation $1/2$. Enforcing the global symmetry of the disorder distribution is necessary to compute the invariant, because we use the subspaces of the symmetry operator to project the

reflection matrix. We present the resulting topological invariant for a single disorder realization in Fig. 5. The invariant correctly changes at the phase transition of the BBH model, even when the lead is attached far from the corner charges. Because scattering invariants rely on the unitarity of the reflection matrix, the phase transition point is sensitive to finite-size effects that may arise from the overlap of the wavefunctions between the inner and outer radius of the annulus. Figures 5(b-f) show the gap of the reflection matrix $\Delta = \min |\det r(\phi)|$ and the invariant $\mathcal{Q}$ as a function of $\lambda/\gamma$ for different outer radii $r$ of the annulus. In the thermodynamic limit of $r \to \infty$ and complete absence of disorder, the invariant changes sign at $\lambda = \gamma$, as expected from the bulk topological invariant of the BBH model. Regardless of the presence of disorder or the size of the annulus, the invariant is always quantized and changes sign only if the reflection matrix has a zero eigenvalue.

## 4.2 Systems with magnetic rotation symmetry

A natural application of scattering invariants arises in network models, which provide an efficient way to compute transport properties of large systems. This makes network models a widely used platform to study localization–delocalization transitions in quantum Hall systems [34,35] and higher-order topological insulators, as done, for example, in Refs. [6,36–38]. Topological invariants in network models, however, have an ambiguous topological classification due to the absence of a bulk Hamiltonian. In this section, we demonstrate how the scattering topological invariant resolves the ambiguous topological classification of HOTI network models by considering a two-dimensional $C_4\mathcal{T}$-symmetric topological superconductor. We use the network model introduced in Ref. [39], where $\mathcal{P}^2 = 1$ and $(C_4\mathcal{T})^4 = -1$, to construct an annulus geometry and, similar to tight-binding models, attach a lead to both the inside and outside of the disk, as shown in Fig. 6(a). We construct the scattering geometry and compute the reflection matrix using a network models package [40].

To construct the scattering invariant, we start by factoring out the $C_2 = (C_4\mathcal{T})^2$ symmetry that commutes with the reflection matrix and with all other symmetries. This consists of projecting the reflection matrix onto one of the two eigensubspaces of $C_2$ and redefining the momentum parameter $\phi := (\phi - \pi)/2$, as described in Sec. 4.1 and App. C. After simplifying

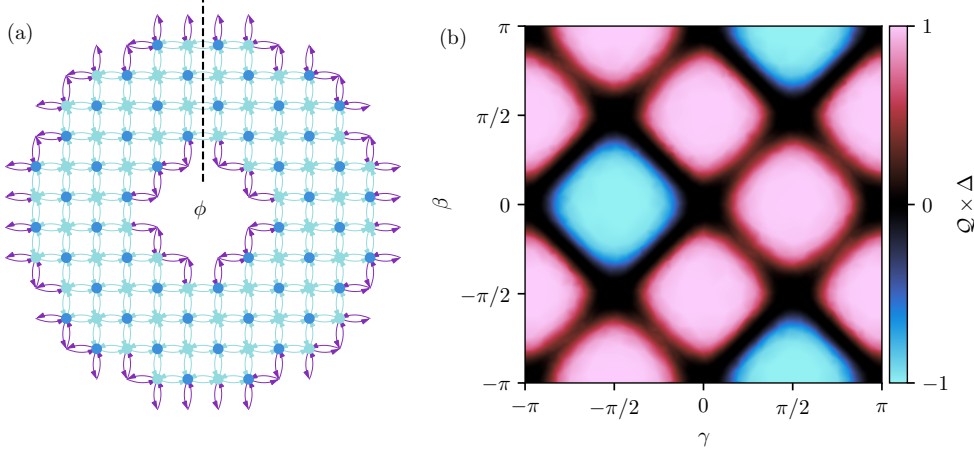

Figure 6: Two-dimensional $C_4\mathcal{T}+\mathcal{P}$-symmetric network model and its corresponding phase diagram. (a) Annulus geometry (blue) with leads (purple) attached to the inside and outside of the disk. The scattering unit cell has two sites and a flux $\phi$ is threaded through its center. (b) Phase diagram as a function of network parameters $\beta, \gamma$ for fixed $\alpha = \pi/2$ and $\delta = 0$.

the symmetry representation, the particle-hole symmetry and the $C_4\mathcal{T}$ symmetry constraints on $r$ become

$$r(\phi) = r^*(-\phi), \qquad r(\phi) = -r^T(-\phi)\exp(-i\phi/2), \tag{15}$$

making the reflection matrix $4\pi$-periodic. At $\phi = 0$, these constraints define a real antisymmetric reflection matrix, and therefore the topological invariant is:

$$\mathcal{Q} = \operatorname{sign}\operatorname{Pf} H_{\text{eff}}(0) = \operatorname{sign}\operatorname{Pf} r(0) \in \mathbb{Z}_2. \tag{16}$$

Similar to the BBH model, the spectral flow of the Hamiltonian spectrum of this model has a pair of zero-modes, which appear at $\phi = 0$, and are protected by particle-hole symmetry and a Kramers-like degeneracy originating from the $C_4\mathcal{T}$ symmetry. Figure 6(b) shows the phase diagram of the network model as a function of the network parameters. The invariant only changes if the reflection matrix has a zero eigenvalue, and it correctly identifies the HOTI phase of the network model as found in Ref. [39].

Our theory for scattering invariants relies on constructing a symmetric scattering geometry and attaching symmetric leads to both the outside and inside surfaces. We further demonstrate the universality of this procedure by applying it to a three-dimensional $C_4\mathcal{T}$-symmetric HOTI [41] with protected hinge modes. We consider the model introduced in Ref. [41]:

$$\begin{aligned}
H(\mathbf{k}) = {} & \tau_z\sigma_0(\cos k_x + \cos k_y + \cos k_z - 3 - \mu) + q_1 \sum_{i=x,y,z} \tau_x\sigma_i \sin k_i \\
& + q_2 \sum_{j=x,y} \tau_y\sigma_j \sin k_j \sin k_z + q_3\tau_x\sigma_0 + p(\cos k_x - \cos k_y)\tau_y\sigma_0,
\end{aligned} \tag{17}$$

where $\sigma_i$ and $\tau_i$ are Pauli matrices acting on the spin and orbital degrees of freedom, $k_i$ are the components of the wave vector, $q_i$ and $p$ are strengths of the Hamiltonian terms that break additional symmetries, and $\mu$ drives the system through the phase transition. Because the model is three-dimensional, the dimensional reduction procedure results in a two-dimensional effective Hamiltonian for the scattering invariant. Additionally, because $C_4\mathcal{T}$ symmetry rotates the system around the $z$-axis, we choose to keep $k_z$ and $\phi$ as the momentum parameters of the effective Hamiltonian, and $x$ and $y$ as the spatial coordinates of the scattering geometry. Therefore, we use a translationally invariant cylindrical geometry with leads attached to the inside and outside of the cylinder, such that a total of eight hinge modes propagate along the outer and inner surfaces, as shown in Fig. 7(a). We construct the scattering geometry and model using Kwant [33] and Qsymm [42].

To construct the scattering invariant, once again we start by factoring out the $C_2 = (C_4\mathcal{T})^2$ symmetry that commutes with the reflection matrix and with all other symmetries, as described in Sec. 4.1. We obtain the reflection matrix constraints as:

$$r(\phi, k_z) = -r^T(-\phi, -k_z)\exp(-i\phi/2). \tag{18}$$

Differently from the two-dimensional network model example, this model does not have a particle-hole symmetry, and the reflection matrix is not constrained to be real. However, at $\phi = 0$ and $k_z = 0, \pi$, the reflection matrix is still real and antisymmetric. The topological invariant is then the same as that of a two-dimensional class AII topological insulator [20], or if expressed in terms of $H_{\text{eff}}$, it is the invariant of the one-dimensional class DIII topological superconductor [43]:

$$\mathcal{Q} = \frac{\operatorname{Pf}[H_{\text{eff}}(0,\pi)]}{\operatorname{Pf}[H_{\text{eff}}(0,0)]} \exp\left[-\frac{1}{2}\int_0^\pi d\log\det H_{\text{eff}}(0,k_z)\right] \in \mathbb{Z}_2, \tag{19}$$

where $H_{\text{eff}}(0, k_z) = r(0, k_z)$. The resulting phase diagram is shown in Fig. 7(b-c).

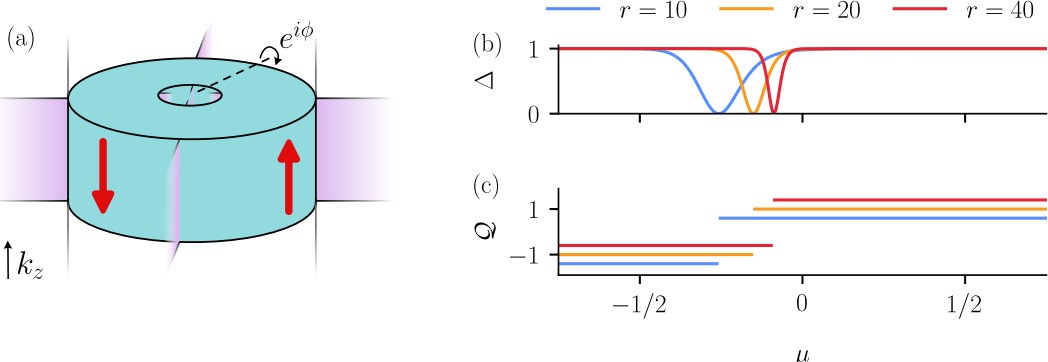

Figure 7: Three-dimensional $C_4\mathcal{T}$-symmetric HOTI. (a) Infinite cylindrical scattering geometry with a flux $\phi$ threaded through the center. The leads (purple) are attached to the inside and outside of the cylinder, both are surfaces with four edge modes (red). (b) Gap of the reflection matrix $\Delta = \min|\det r(\phi, k_z)|$ and (c) invariant $\mathcal{Q}$ as a function of the onsite potential $\mu$. For this computation we use $q_1 = 1$ and $q_2 = q_3 = p = 0.1$.

We observe that both in the two-dimensional and three-dimensional cases, the topological invariant is determined by the reflection matrix at $\phi = 0$. This is similar to an earlier work on antiferromagnetic topological insulators [44]. Such behavior occurs whenever an antiunitary symmetry is multiplied by a fractional translation, which makes the square of the symmetry operator change between different high symmetry lines in the Brillouin zone. In our construction, the same principle applies to fractional rotations. Thanks to this observation, we also conclude that the topological classification of systems with magnetic rotation exponentiating to $+1$ is identical, except that the relevant value of $\phi$ shifts in the reduced Brillouin zone.

## 4.3 Axion insulator: 3D inversion

Finally, in this section, we demonstrate how to apply the scattering invariant to point-group symmetries, which lack a preferred direction for the leads or boundary modes. As an example, we consider a three-dimensional axion insulator [45–47], which is a second-order HOTI protected by inversion symmetry. A finite axion insulator sample hosts a chiral mode that encircles the entire sample [48], as shown in Fig. 8(a). Inversion symmetry forbids the chiral mode from being contracted to a point, but it does not constrain its position on the surface.

We construct a minimal model of an axion insulator with the following Hamiltonian:

$$H = -\mu\tau_z\sigma_0 + \sum_{i=x,y,z} 2(1-\cos k_i)\tau_z\sigma_0 + \alpha \sum_{i=x,y,z} \sin k_i\sigma_i\tau_x + \boldsymbol{B}\tau_0\boldsymbol{\sigma} \,. \tag{20}$$

Here $\sigma_i$ and $\tau_i$ are Pauli matrices acting on the spin and orbital degrees of freedom, and $k_i$ are the components of the wave vector. The Hamiltonian parameter $\alpha$ is the spin-orbit coupling strength, $\boldsymbol{B}$ is a magnetic field, which we use to break time-reversal symmetry, and $\mu$ is the chemical potential. This model is trivial for $\mu < 0$ and topological for $\mu > 0$. Because the HOTI is protected by inversion symmetry, we construct an inversion-symmetric scattering geometry for applying the scattering invariant. Such a geometry is a translationally-invariant cylinder where $k_z$ and $\phi$ are the momentum parameters of the effective Hamiltonian, and $x$ and $y$ are the spatial coordinates of the scattering geometry.

Inversion symmetry maps $k_z \to -k_z$ and $\phi \to \phi$, and constrains the reflection matrix $r(\phi, k_z)$ as:

$$V_\mathcal{I}(\phi)r(\phi, k_z)Q_\mathcal{I}^\dagger(\phi) = r(\phi, -k_z), \qquad V_\mathcal{I}^2(\phi) = Q_\mathcal{I}^2(\phi) = e^{-i\phi} \,. \tag{21}$$

This follows from Eqs. (5a) and (21), and becomes a commutation relation for $k_z = 0, \pi$. The equivalent $H_{\text{eff}}$ (3) is two-dimensional, and it has a combination of a glide symmetry and a sublattice symmetry. This topological phase was analyzed in Ref. [49], and its topological invariant relies on block-diagonalizing the reflection matrix by projecting it onto the eigenbasis of $V_{\mathcal{I}}(\phi)$ and $Q_{\mathcal{I}}(\phi)$ at $k_z = 0$ and $k_z = \pi$. The scattering invariant is:

$$\mathcal{Q} = \text{sign}\left[\frac{\det r_+(-\pi, 0)}{\det r_+(-\pi, \pi)} \frac{\exp \frac{1}{2} \int_{-\pi}^{\pi} d_\phi \ln \det r_+(\phi, 0)}{\exp \frac{1}{2} \int_{-\pi}^{\pi} d_\phi \ln \det r_+(\phi, \pi)} \exp \frac{1}{2} \int_0^\pi d_{k_z} \ln \det r(-\pi, k_z)\right], \quad (22)$$

where $r_+(\phi, k_z)$ is the reflection matrix projected onto the subspace of $V_{\mathcal{I}}(\phi)$ and $Q_{\mathcal{I}}(\phi)$ with eigenvalue $e^{i\phi/2}$. Its graphical representation is shown in Fig. 8(b), and we identify the numerator and denominator of the fraction as the way of defining $\sqrt{\det r(-\pi, 0)}$ and $\sqrt{\det r(-\pi, \pi)}$ without the sign ambiguity. Similar to Sec. 4.1, we apply the procedure of App. D to ensure that the integrands are smooth functions of $\phi$. By applying the invariant to the scattering matrix of our example system, we confirm that the invariant switches at the phase transition, as shown in Fig. 8(c-d). Fig. 8(d) shows an extended range of $\mu$ where the invariant flips sign due to the Fermi surface that appears for such values of $\mu$.

Scattering invariants are a powerful tool to study localization–delocalization transitions in disordered systems because they establish a direct connection between topology and transport properties. Reference [6], for example, studied the effect of the bulk delocalization transition of disordered axion insulators using a network model, and claimed that an earlier analysis [50, 51] of topological phases protected by an average symmetry does not extend to HOTIs. Our scattering invariant, however, allows us to formulate a proof of the existence of the delocalized phase in a disordered axion insulator along the lines of Ref. [50]. We do this by considering a finite but large cylindrical sample with disorder in one half being an inversion image of the disorder in the other half, such that locally disorder breaks inversion symmetry, but globally the system is inversion symmetric. Because the scattering invariant must change across the phase transition, a mode must perfectly transmit between the inner and outer radius of the cylinder as a phase transition is crossed. Additionally, because the system is three-dimensional, this implies that the perfectly transmitted mode is accompanied by a metallic phase. This argument establishes that the axion insulator must have a delocalized phase in the presence of disorder that is symmetric on average.

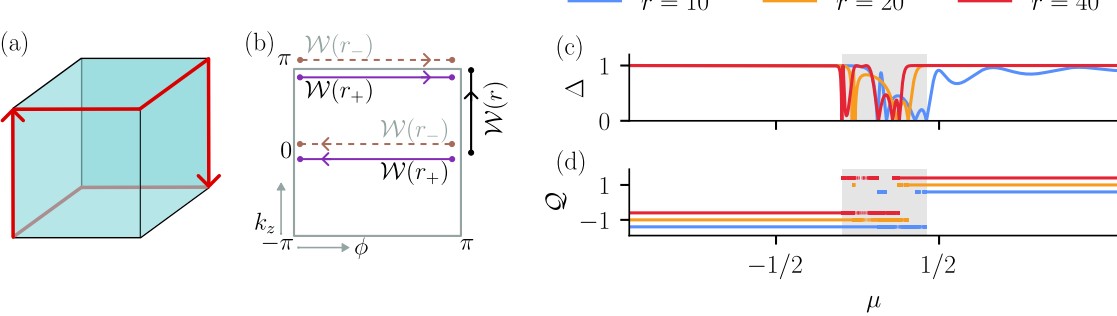

Figure 8: A three-dimensional inversion-symmetric second-order HOTI, the axion insulator. (a) Finite geometry in the HOTI phase with a protected edge mode (red) encircling the entire sample. (b) Brillouin zone of the effective Hamiltonian $H_{\text{eff}}$ with the path and windings $\mathcal{W}$ of the invariant calculation. (c) Gap of the reflection matrix $\Delta = \min|\det r(\phi, k_z)|$. (d) Scattering invariant $\mathcal{Q}$ as a function of the chemical potential $\mu$. The extended range of $\mu$ (gray) for which the sign of the invariant changes is due to the Fermi surface.

# 5 Conclusion and discussion

We presented a general construction of scattering invariants for second order intrinsic higher order topological insulators and superconductors. To do so, we established a general procedure that relies on the compatibility of the spatial symmetries that protect a HOTI with the geometry of the scattering region. Determining the topological invariant required us to consider a geometry with separated inside and outside regions, and to consider the flux dependence of the reflection matrix. Similar to the dimensional reduction procedure for strong topological insulators [20], from a given model we constructed a lower-dimensional effective Hamiltonian in a different symmetry class, and used the topological invariant of this effective Hamiltonian to determine the topological phase of the original model. Differently from the standard scattering theory of topological insulators, our procedure maps a rotation symmetry on a translation symmetry, and an inversion symmetry on a glide symmetry.

We have applied the procedure to several prominent examples:

- The BBH model, a 2D HOTI protected by anticommuting $C_4$ and $\mathcal{C}$ symmetries.

- A 2D HOTI with a $C_4\mathcal{T}$ and $\mathcal{P}$ symmetry, for which we used a network model demonstration.

- A 3D HOTI with $C_4\mathcal{T}$ symmetry.

- A 3D axion insulator with inversion symmetry.

In all cases, the scattering invariant correctly identified the topological phase transitions regardless of the detailed geometry of the scattering region. The topological invariants that we used employ reasoning previously identified for antiferromagnetic topological insulators [44] or for topological crystalline insulators [49]. Additionally, our approach enables us to factor out a pure rotation that commutes with the other symmetries, while the Hamiltonian approach [41] requires special treatment of such symmetries.

Our examples, together with general arguments, indicate that our approach applies to all the second order HOTIs. On the other hand, higher order HOTIs, in particular three-dimensional phases with zero-dimensional protected modes, do not seem to fit into our framework. This is because one can choose the cylinder geometry so that the corner modes do not appear on its inner or outer boundary. We expect that the need for holes with a flux in the scattering geometry extends to higher order HOTIs. More specifically, we conjecture that third order HOTIs need a scattering geometry with two holes and two fluxes $\phi_i$ threaded through them, in such a way that the dimensional reduction procedure maps to a $d - 1$-dimensional $H_{\text{eff}}(\phi_1, \phi_2, \boldsymbol{k}_{d-3})$. The finite size gap of the surface states at the boundaries then goes to zero as a function of the fluxes introduced.

Finally, our work reveals a different perspective on higher order topological insulators from those presented in the literature. We have found that a nontrivial reflection matrix always corresponds to the appearance of a zero energy mode in the original model's spectrum, as exemplified in the spectral flow of Fig. 4. Because the topology of $r$ may not change when the geometry is varied smoothly, we may contract the inner radius of the cylinder into a point and extend the outer radius to infinity. This demonstrates that all second order HOTIs have protected defect modes appearing at flux lines, which is a local property that was previously overlooked. It is interesting to explore whether this property characterizes disordered higher order topological phases, like those studied in Ref. [8]. It is also interesting to apply our framework to identify quantized transport phenomena in HOTIs, likely related to the interplay between adiabatic pumping [52, 53] and symmetry eigenvalues.

## Acknowledgments

We thank Cosma Fulga, Hui Liu, and Dániel Varjas for sharing unpublished results and multiple useful discussions. We also thank Adam Chaou and Hélène Spring for useful discussions.

**Data availability**    The code used to produce the reported results is available on Zenodo [54]. Additionally, we used a Python package for network models that is currently under development [40].

**Author contributions**    All authors contributed to the initial idea, final project scope, identification of the topological invariants, writing code, and writing the manuscript. A. A. oversaw the project.

**Funding information**    This research was supported by the Netherlands Organization for Scientific Research (NWO/OCW) as part of the Frontiers of Nanoscience program, and an NWO VIDI grant 016.Vidi.189.180.

## A    Scattering equation

The scattering formalism describes the transport of electrons through a scattering region connected to leads. The scattering region is a finite system of sites and orbitals that hosts localized wavefunctions, while the leads are semi-infinite and translational invariant, and thus host plane waves. On the basis of sites and orbitals of the scattering region and the leads, the Hamiltonian of the system is:

$$
H = \begin{pmatrix}
H_{\mathrm{sr}} & P_{\mathrm{sr}}^T V^\dagger & & & \\
V P_{\mathrm{sr}} & H_{\mathrm{lead}} & V^\dagger & & \\
& V & H_{\mathrm{lead}} & V^\dagger & \\
& & V & H_{\mathrm{lead}} & V^\dagger \\
& & & \ddots & \ddots & \ddots
\end{pmatrix},
\tag{A.1}
$$

where $H_{\mathrm{sr}}$ is the (big) Hamiltonian of the scattering region and $H_{\mathrm{lead}}$ is the (small) Hamiltonian of the lead's unit cell. Here we grouped all the leads into a single one. The coupling of the lead to the scattering region takes the specific form $V P_{\mathrm{sr}}$ where $V$ is the hopping matrix between the leads and the scattering region and $P_{\mathrm{sr}}$ is the projector $P_{\mathrm{sr}}$ rectangular matrix with ones in the diagonal and zeros everywhere else. Reference [22] contains a detailed review of the scattering matrix formalism.

The scattering matrix $S(E)$ relates the incoming and outgoing wavefunctions between the leads at an energy $E$, and its elements are given by the scattering equations:

$$
(H_{ab} - E\delta_{ab})\left( \Psi_{bn}^{\mathrm{in}} \alpha_n^{\mathrm{in}} + \sum_m \Psi_{bm}^{\mathrm{out}} S_{mn} \alpha_n^{\mathrm{in}} + \Psi_{bn}^{\mathrm{localized}} \alpha_n^{\mathrm{in}} \right) = 0\,,
\tag{A.2}
$$

where $H_{ab}$ is the Hamiltonian, and $a$ and $b$ label the sites and orbital degrees of freedom of the scattering region and the leads. The incoming and outgoing wavefunctions $\Psi^{\mathrm{in}}$ and $\Psi^{\mathrm{out}}$ are finite in the leads and vanish in the scattering region, while the localized wavefunctions $\Psi^{\mathrm{localized}}$ are defined in the scattering region and may decay exponentially into the leads if evanescent modes are present. The wavefunctions are matrices where each column corresponds to a mode, and the rows label the sites and orbitals matching the size of the Hamiltonian. The

total number of modes is $N_{\mathrm{modes}} = N_{\mathrm{sites}} \times N_{\mathrm{orbitals}}$ for ideal leads, i.e., leads that do not host any evanescence modes, where $N_{\mathrm{sites}}$ is the number of sites from which the leads are attached, and $N_{\mathrm{orbitals}}$ is the number of orbitals per site in the lead. The scalars $\alpha_n^{\mathrm{in}}$ are the amplitudes of each incoming mode $n$. Because the leads are translational-invariant, the leads modes are plane waves and we solve Eq. (A.2) using a sparse finite system of equations [22].

In the context of scattering topological invariants, we are interested in the topology of the scattering region. Therefore, $H_{\mathrm{sr}}$ is a HOTI Hamiltonian that preserves the symmetries of interest, while $H_{\mathrm{lead}}$ only needs to globally preserve the symmetries. In practice, we create either a tight-binding model of the scattering region or a network model. We make the tight-binding model of a scattering region using the Kwant package [33] and test the symmetries of the Hamiltonian using Qsymm [42]. Additionally, we construct ideal leads using Kwant, see Ref. [54] for the code. A network model, such as we used for the 2D $C_4\mathcal{T} + \mathcal{P}$ symmetric system in section 4.2 consists of a combination of scattering matrices, each as a node within the network. The links in between the nodes connect the different scattering regions and the leads are once again plane waves to the scattering regions.

# B Symmetry constraints on scattering matrices

In this section, we derive the constraints on the reflection matrix $r$ imposed by a symmetry operator $\mathcal{O}$. In general, applying an operator $\mathcal{O}$ to the scattering equation in Eq. (1) at $E = 0$ gives:

$$\tilde{H}(R_{\mathcal{O}}(\phi, \boldsymbol{k}_{\mathrm{d\text{-}2}}))\left(\mathcal{O}\Psi^{\mathrm{in}}\alpha^{\mathrm{in}} + \mathcal{O}\Psi^{\mathrm{out}}S(\phi, \boldsymbol{k}_{\mathrm{d\text{-}2}})\alpha^{\mathrm{in}} + \mathcal{O}\Psi^{\mathrm{localized}}\alpha^{\mathrm{in}}\right) = 0, \tag{B.1}$$

where $\tilde{H}(R_{\mathcal{O}}(\phi, \boldsymbol{k}_{\mathrm{d\text{-}2}})) = \mathcal{O}H(\phi, \boldsymbol{k}_{\mathrm{d\text{-}2}})\mathcal{O}^{-1}$ is the transformed Hamiltonian, and $R_{\mathcal{O}}$ is the action of the operator in parameter space. If the operator $\mathcal{O}$ is a symmetry of the Hamiltonian, Eq. (B.1) constrains the scattering matrix $S$ and the reflection matrix $r$. To derive the constraints on $r$, we equate the coefficients of the wavefunctions in the scattering equations to the ones in the symmetry-transformed scattering equations. Therefore, we start by applying $\mathcal{O}$ to the incoming and outgoing wavefunctions. Because $\mathcal{O}$ may be a unitary or antiunitary operator, and a symmetry or antisymmetry, we consider four cases.

## B.1 Unitary symmetry

Unitary symmetries map incoming and outgoing wavefunctions within the same Hilbert space:

$$\mathcal{O}\Psi^{\mathrm{in}} = \Psi^{\mathrm{in}}V_{\mathcal{O}}(\phi, \boldsymbol{k}_{\mathrm{d\text{-}2}}), \qquad \mathcal{O}\Psi^{\mathrm{out}} = \Psi^{\mathrm{out}}Q_{\mathcal{O}}(\phi, \boldsymbol{k}_{\mathrm{d\text{-}2}}), \tag{B.2}$$

where $V_{\mathcal{O}}$ and $Q_{\mathcal{O}}$ are matrices that act on the wavefunctions. Applying $\mathcal{O}$ to the wavefunctions twice gives:

$$\begin{aligned}
\mathcal{O}(\mathcal{O}\Psi^{\mathrm{in}}) &= \mathcal{O}(\Psi^{\mathrm{in}}V_{\mathcal{O}}) = \Psi^{\mathrm{in}}V_{\mathcal{O}}^2(\phi, \boldsymbol{k}_{\mathrm{d\text{-}2}}), \\
\mathcal{O}(\mathcal{O}\Psi^{\mathrm{out}}) &= \mathcal{O}(\Psi^{\mathrm{out}}Q_{\mathcal{O}}) = \Psi^{\mathrm{in}}Q_{\mathcal{O}}^2(\phi, \boldsymbol{k}_{\mathrm{d\text{-}2}}).
\end{aligned} \tag{B.3}$$

As a consequence, we obtain the following constraint:

$$\begin{aligned}
(\Psi^{\mathrm{in}})^{\dagger}\mathcal{O}^2\Psi^{\mathrm{in}} &= V_{\mathcal{O}}^2(\phi, \boldsymbol{k}_{\mathrm{d\text{-}2}}), \\
(\Psi^{\mathrm{out}})^{\dagger}\mathcal{O}^2\Psi^{\mathrm{out}} &= Q_{\mathcal{O}}^2(\phi, \boldsymbol{k}_{\mathrm{d\text{-}2}}),
\end{aligned} \tag{B.4}$$

if $\mathcal{O}$ is a unitary symmetry.

Finally, we combine Eqs. (B.2) and (B.1):

$$\tilde{H}(R_{\mathcal{O}}(\phi, \boldsymbol{k}_{\mathrm{d\text{-}2}}))\left(\Psi^{\mathrm{in}}V_{\mathcal{O}}(\phi, \boldsymbol{k}_{\mathrm{d\text{-}2}})\alpha^{\mathrm{in}} + \Psi^{\mathrm{out}}Q_{\mathcal{O}}(\phi, \boldsymbol{k}_{\mathrm{d\text{-}2}})S(\phi, \boldsymbol{k}_{\mathrm{d\text{-}2}})\alpha^{\mathrm{in}} + \Psi^{\mathrm{localized}}\alpha^{\mathrm{in}}\right) = 0 \tag{B.5}$$

$$\implies \tilde{H}(R_{\mathcal{O}}(\phi, \boldsymbol{k}_{\mathrm{d\text{-}2}}))\left(\Psi^{\mathrm{in}}\alpha^{\mathrm{in}\prime} + \Psi^{\mathrm{out}}Q_{\mathcal{O}}(\phi, \boldsymbol{k}_{\mathrm{d\text{-}2}})S(\phi, \boldsymbol{k}_{\mathrm{d\text{-}2}})V_{\mathcal{O}}^{\dagger}(\phi, \boldsymbol{k}_{\mathrm{d\text{-}2}})\alpha^{\mathrm{in}\prime} + \Psi^{\mathrm{localized}}\alpha^{\mathrm{in}}\right) = 0,$$

where we identified $\alpha^{\mathrm{in}'} = V_{\mathcal{O}}(\phi, \boldsymbol{k}_{\text{d-2}})\alpha^{\mathrm{in}}$. Because $\mathcal{O}$ is a symmetry, the solutions of Eq. (B.5) are those of Eq. (1) for $S(R_{\mathcal{O}}(\phi, \boldsymbol{k}_{\text{d-2}}))$. Therefore, the scattering matrix $S$ is constrained by the symmetry operator $\mathcal{O}$ as:

$$Q_{\mathcal{O}}(\phi, \boldsymbol{k}_{\text{d-2}})S(\phi, \boldsymbol{k}_{\text{d-2}})V_{\mathcal{O}}^{\dagger}(\phi, \boldsymbol{k}_{\text{d-2}}) = S(R_{\mathcal{O}}(\phi, \boldsymbol{k}_{\text{d-2}})). \tag{B.6}$$

## B.2 Antiunitary antisymmetry

Antiunitary antisymmetries map incoming and outgoing wavefunctions within the same Hilbert space, like unitary symmetries:

$$\mathcal{O}\Psi^{\mathrm{in}} = \Psi^{\mathrm{in}}V_{\mathcal{O}}(\phi, \boldsymbol{k}_{\text{d-2}}), \qquad \mathcal{O}\Psi^{\mathrm{out}} = \Psi^{\mathrm{out}}Q_{\mathcal{O}}(\phi, \boldsymbol{k}_{\text{d-2}}), \tag{B.7}$$

where $V_{\mathcal{O}}$ and $Q_{\mathcal{O}}$ are matrices that act on the wavefunctions. Applying $\mathcal{O}$ to the wavefunctions twice gives:

$$\begin{aligned} \mathcal{O}(\mathcal{O}\Psi^{\mathrm{in}}) &= \mathcal{O}(\Psi^{\mathrm{in}}V_{\mathcal{O}}) = \Psi^{\mathrm{in}}V_{\mathcal{O}}(\phi, \boldsymbol{k}_{\text{d-2}})V_{\mathcal{O}}^{*}(\phi, \boldsymbol{k}_{\text{d-2}}), \\ \mathcal{O}(\mathcal{O}\Psi^{\mathrm{out}}) &= \mathcal{O}(\Psi^{\mathrm{out}}Q_{\mathcal{O}}) = \Psi^{\mathrm{in}}Q_{\mathcal{O}}(\phi, \boldsymbol{k}_{\text{d-2}})Q_{\mathcal{O}}^{*}(\phi, \boldsymbol{k}_{\text{d-2}}), \end{aligned} \tag{B.8}$$

where we applied the conjugate operator $\mathcal{K}$ to the matrices $V_{\mathcal{O}}$ and $Q_{\mathcal{O}}$. As a consequence, we obtain the following constraint:

$$\begin{aligned} (\Psi^{\mathrm{in}})^{\dagger}\mathcal{O}^2\Psi^{\mathrm{in}} &= V_{\mathcal{O}}(\phi, \boldsymbol{k}_{\text{d-2}})V_{\mathcal{O}}^{*}(\phi, \boldsymbol{k}_{\text{d-2}}), \\ (\Psi^{\mathrm{out}})^{\dagger}\mathcal{O}^2\Psi^{\mathrm{out}} &= Q_{\mathcal{O}}(\phi, \boldsymbol{k}_{\text{d-2}})Q_{\mathcal{O}}^{*}(\phi, \boldsymbol{k}_{\text{d-2}}), \end{aligned} \tag{B.9}$$

if $\mathcal{O}$ is an antiunitary antisymmetry.

Finally, we combine Eqs. (B.7) and (B.1):

$$\tilde{H}(R_{\mathcal{O}}(\phi, \boldsymbol{k}_{\text{d-2}}))\big(\Psi^{\mathrm{in}}V_{\mathcal{O}}(\phi, \boldsymbol{k}_{\text{d-2}})\alpha^{\mathrm{in}*} + \Psi^{\mathrm{out}}Q_{\mathcal{O}}(\phi, \boldsymbol{k}_{\text{d-2}})S^{*}(\phi, \boldsymbol{k}_{\text{d-2}})\alpha^{\mathrm{in}*} + \Psi^{\mathrm{localized}}\alpha^{\mathrm{in}*}\big) = 0 \tag{B.10}$$

$$\implies \tilde{H}(R_{\mathcal{O}}(\phi, \boldsymbol{k}_{\text{d-2}}))\big(\Psi^{\mathrm{in}}\alpha^{\mathrm{in}'} + \Psi^{\mathrm{out}}Q_{\mathcal{O}}(\phi, \boldsymbol{k}_{\text{d-2}})S^{*}(\phi, \boldsymbol{k}_{\text{d-2}})V_{\mathcal{O}}^{\dagger}(\phi, \boldsymbol{k}_{\text{d-2}})\alpha^{\mathrm{in}'} + \Psi^{\mathrm{localized}}\alpha^{\mathrm{in}}\big) = 0,$$

where we identified $\alpha^{\mathrm{in}'} = V_{\mathcal{O}}(\phi, \boldsymbol{k}_{\text{d-2}})\alpha^{\mathrm{in}*}$. Because $\mathcal{O}$ is a symmetry, the solutions of Eq. (B.10) are those of Eq. (1) for $S(R_{\mathcal{O}}(\phi, \boldsymbol{k}_{\text{d-2}}))$. Therefore, the scattering matrix $S$ is constrained by the symmetry operator $\mathcal{O}$ as:

$$Q_{\mathcal{O}}(\phi, \boldsymbol{k}_{\text{d-2}})S^{*}(\phi, \boldsymbol{k}_{\text{d-2}})V_{\mathcal{O}}^{\dagger}(\phi, \boldsymbol{k}_{\text{d-2}}) = S(R_{\mathcal{O}}(\phi, \boldsymbol{k}_{\text{d-2}})). \tag{B.11}$$

For example, in the presence of particle-hole symmetry Eq. (B.11) establishes that $S$ is isomorphic to a real matrix $S'$ at time-reversal invariant momenta if $\mathcal{P}^2 = 1$. To see this, we use that $V_{\mathcal{P}}V_{\mathcal{P}}^{*} = Q_{\mathcal{P}}Q_{\mathcal{P}}^{*} = \mathcal{P}^2 = \pm 1$ to manipulate Eq.(B.11) and obtain

$$\left(\sqrt{Q_{\mathcal{P}}(\phi, \boldsymbol{k}_{\text{d-2}})}^{*}S(\phi, \boldsymbol{k}_{\text{d-2}})\sqrt{V_{\mathcal{P}}(\phi, \boldsymbol{k}_{\text{d-2}})}^{T}\right)^{*} = \pm\sqrt{Q_{\mathcal{P}}(\phi, \boldsymbol{k}_{\text{d-2}})}^{*}S(-\phi, -\boldsymbol{k}_{\text{d-2}})\sqrt{V_{\mathcal{P}}(\phi, \boldsymbol{k}_{\text{d-2}})}^{T}, \tag{B.12}$$

where $\sqrt{Q_{\mathcal{P}}}$ and $\sqrt{V_{\mathcal{P}}}$ denote the square root of the matrix. At time-reversal invariant momenta we define $S' = \sqrt{Q_{\mathcal{P}}}^{*}S\sqrt{V_{\mathcal{P}}^{T}}$ and find $S' = S'^{*}$ for $\mathcal{P}^2 = 1$, and $S' = \sigma_y S'^{*}\sigma_y$ for $\mathcal{P}^2 = -1$. The $\sigma_y$ operator is required to satisfy $\sqrt{Q_{\mathcal{P}}}\sqrt{Q_{\mathcal{P}}^{*}} = \sqrt{V_{\mathcal{P}}}\sqrt{V_{\mathcal{P}}^{*}} = -1$. This constraint may be incompatible with other constraints if multiple symmetries are present.

## B.3 Unitary antisymmetry

Unitary antisymmetries map incoming and outgoing wavefunctions to each other:

$$\mathcal{O}\Psi^{\mathrm{in}} = \Psi^{\mathrm{out}}V_{\mathcal{O}}(\phi, \boldsymbol{k}_{\text{d-2}}), \qquad \mathcal{O}\Psi^{\mathrm{out}} = \Psi^{\mathrm{in}}Q_{\mathcal{O}}(\phi, \boldsymbol{k}_{\text{d-2}}), \tag{B.13}$$

where $V_{\mathcal{O}}$ and $Q_{\mathcal{O}}$ are matrices that act on the wavefunctions. Applying $\mathcal{O}$ to the wavefunctions twice gives:

$$
\begin{aligned}
\mathcal{O}(\mathcal{O}\Psi^{\text{in}}) &= \mathcal{O}(\Psi^{\text{out}}V_{\mathcal{O}}) = \Psi^{\text{in}}Q_{\mathcal{O}}(\phi,\boldsymbol{k}_{\text{d-2}})V_{\mathcal{O}}(\phi,\boldsymbol{k}_{\text{d-2}}), \\
\mathcal{O}(\mathcal{O}\Psi^{\text{out}}) &= \mathcal{O}(\Psi^{\text{in}}Q_{\mathcal{O}}) = \Psi^{\text{in}}V_{\mathcal{O}}(\phi,\boldsymbol{k}_{\text{d-2}})Q_{\mathcal{O}}(\phi,\boldsymbol{k}_{\text{d-2}}),
\end{aligned}
\tag{B.14}
$$

where we applied the conjugate operator $\mathcal{K}$ to the matrices $V_{\mathcal{O}}$ and $Q_{\mathcal{O}}$. As a consequence, we obtain the following constraint:

$$
\begin{aligned}
(\Psi^{\text{in}})^{\dagger}\mathcal{O}^2\Psi^{\text{in}} &= Q_{\mathcal{O}}(\phi,\boldsymbol{k}_{\text{d-2}})V_{\mathcal{O}}(\phi,\boldsymbol{k}_{\text{d-2}}), \\
(\Psi^{\text{out}})^{\dagger}\mathcal{O}^2\Psi^{\text{out}} &= V_{\mathcal{O}}(\phi,\boldsymbol{k}_{\text{d-2}})Q_{\mathcal{O}}(\phi,\boldsymbol{k}_{\text{d-2}}),
\end{aligned}
\tag{B.15}
$$

if $\mathcal{O}$ is a unitary antisymmetry.

Finally, we combine Eqs. (B.13) and (B.1):

$$
\tilde{H}(R_{\mathcal{O}}(\phi,\boldsymbol{k}_{\text{d-2}}))\left(\Psi^{\text{out}}V_{\mathcal{O}}(\phi,\boldsymbol{k}_{\text{d-2}})\alpha^{\text{in}} + \Psi^{\text{in}}Q_{\mathcal{O}}(\phi,\boldsymbol{k}_{\text{d-2}})S(\phi,\boldsymbol{k}_{\text{d-2}})\alpha^{\text{in}} + \Psi^{\text{localized}}\alpha^{\text{in}}\right) = 0 \tag{B.16}
$$

$$
\implies \quad \tilde{H}(R_{\mathcal{O}}(\phi,\boldsymbol{k}_{\text{d-2}}))\left(\Psi^{\text{in}}\alpha^{\text{in}'} + \Psi^{\text{out}}V_{\mathcal{O}}(\phi,\boldsymbol{k}_{\text{d-2}})S^{\dagger}(\phi,\boldsymbol{k}_{\text{d-2}})Q_{\mathcal{O}}^{\dagger}(\phi,\boldsymbol{k}_{\text{d-2}})\alpha^{\text{in}'} + \Psi^{\text{localized}}\alpha^{\text{in}}\right) = 0,
$$

where we identified $\alpha^{\text{in}'} = Q_{\mathcal{O}}(\phi,\boldsymbol{k}_{\text{d-2}})S(\phi,\boldsymbol{k}_{\text{d-2}})\alpha^{\text{in}}$. Because $\mathcal{O}$ is a symmetry, the solutions of Eq. (B.16) are those of Eq. (1) for $S(R_{\mathcal{O}}(\phi,\boldsymbol{k}_{\text{d-2}}))$. Therefore, the scattering matrix $S$ is constrained by the symmetry operator $\mathcal{O}$ as:

$$
V_{\mathcal{O}}(\phi,\boldsymbol{k}_{\text{d-2}})S^{\dagger}(\phi,\boldsymbol{k}_{\text{d-2}})Q_{\mathcal{O}}^{\dagger}(\phi,\boldsymbol{k}_{\text{d-2}}) = S(R_{\mathcal{O}}(\phi,\boldsymbol{k}_{\text{d-2}})). \tag{B.17}
$$

For example, in the presence of chiral symmetry Eq. (B.17) establishes that $S$ is isomorphic to a Hermitian matrix $S'$. To find the transformation that makes $S$ Hermitian, we first observe that $Q_{\mathcal{C}}V_{\mathcal{C}} = V_{\mathcal{C}}Q_{\mathcal{C}} = \mathcal{C}^2 = 1$ from Eq. (B.15). Then, we use $Q_{\mathcal{C}}^{\dagger} = V_{\mathcal{C}}$ in Eq. (B.17) and find $S' = S'^{\dagger}$ for $S'(\phi,\boldsymbol{k}_{\text{d-2}}) = V_{\mathcal{C}}^{\dagger}(\phi,\boldsymbol{k}_{\text{d-2}})S(\phi,\boldsymbol{k}_{\text{d-2}})$.

## B.4 Antiunitary symmetry

Antiunitary symmetries map incoming and outgoing wavefunctions to each other:

$$
\mathcal{O}\Psi^{\text{in}} = \Psi^{\text{out}}V_{\mathcal{O}}(\phi,\boldsymbol{k}_{\text{d-2}}), \qquad \mathcal{O}\Psi^{\text{out}} = \Psi^{\text{in}}Q_{\mathcal{O}}(\phi,\boldsymbol{k}_{\text{d-2}}), \tag{B.18}
$$

where $V_{\mathcal{O}}$ and $Q_{\mathcal{O}}$ are matrices that act on the wavefunctions. Applying $\mathcal{O}$ to the wavefunctions twice gives:

$$
\begin{aligned}
\mathcal{O}(\mathcal{O}\Psi^{\text{in}}) &= \mathcal{O}(\Psi^{\text{out}}V_{\mathcal{O}}) = \Psi^{\text{in}}Q_{\mathcal{O}}(\phi,\boldsymbol{k}_{\text{d-2}})V_{\mathcal{O}}^*(\phi,\boldsymbol{k}_{\text{d-2}}), \\
\mathcal{O}(\mathcal{O}\Psi^{\text{out}}) &= \mathcal{O}(\Psi^{\text{in}}Q_{\mathcal{O}}) = \Psi^{\text{in}}V_{\mathcal{O}}(\phi,\boldsymbol{k}_{\text{d-2}})Q_{\mathcal{O}}^*(\phi,\boldsymbol{k}_{\text{d-2}}),
\end{aligned}
\tag{B.19}
$$

where we applied the conjugate operator $\mathcal{K}$ to the matrices $V_{\mathcal{O}}$ and $Q_{\mathcal{O}}$. As a consequence, we obtain the following constraint:

$$
\begin{aligned}
(\Psi^{\text{in}})^{\dagger}\mathcal{O}^2\Psi^{\text{in}} &= Q_{\mathcal{O}}(\phi,\boldsymbol{k}_{\text{d-2}})V_{\mathcal{O}}^*(\phi,\boldsymbol{k}_{\text{d-2}}), \\
(\Psi^{\text{out}})^{\dagger}\mathcal{O}^2\Psi^{\text{out}} &= V_{\mathcal{O}}(\phi,\boldsymbol{k}_{\text{d-2}})Q_{\mathcal{O}}^*(\phi,\boldsymbol{k}_{\text{d-2}}),
\end{aligned}
\tag{B.20}
$$

if $\mathcal{O}$ is an antiunitary symmetry.

Finally, we combine Eqs. (B.18) and (B.1):

$$
\tilde{H}(R_{\mathcal{O}}(\phi,\boldsymbol{k}_{\text{d-2}}))\left(\Psi^{\text{out}}V_{\mathcal{O}}(\phi,\boldsymbol{k}_{\text{d-2}})\alpha^{\text{in}*} + \Psi^{\text{in}}Q_{\mathcal{O}}(\phi,\boldsymbol{k}_{\text{d-2}})S^*(\phi,\boldsymbol{k}_{\text{d-2}})\alpha^{\text{in}*} + \Psi^{\text{localized}}\alpha^{\text{in}*}\right) = 0 \tag{B.21}
$$

$$
\implies \quad \tilde{H}(R_{\mathcal{O}}(\phi,\boldsymbol{k}_{\text{d-2}}))\left(\Psi^{\text{in}}\alpha^{\text{in}'} + \Psi^{\text{out}}V_{\mathcal{O}}(\phi,\boldsymbol{k}_{\text{d-2}})S^T(\phi,\boldsymbol{k}_{\text{d-2}})Q_{\mathcal{O}}^{\dagger}(\phi,\boldsymbol{k}_{\text{d-2}})\alpha^{\text{in}'} + \Psi^{\text{localized}}\alpha^{\text{in}}\right) = 0,
$$

where we identified $\alpha^{\text{in}'} = Q_{\mathcal{O}}(\phi, \boldsymbol{k}_{\text{d-2}}) S^*(\phi, \boldsymbol{k}_{\text{d-2}}) \alpha^{\text{in}*}$. Because $\mathcal{O}$ is a symmetry, the solutions of Eq. (B.21) are those of Eq. (1) for $S(R_{\mathcal{O}}(\phi, \boldsymbol{k}_{\text{d-2}}))$. Therefore, the scattering matrix $S$ is constrained by the symmetry operator $\mathcal{O}$ as:

$$V_{\mathcal{O}}(\phi, \boldsymbol{k}_{\text{d-2}}) S^T(\phi, \boldsymbol{k}_{\text{d-2}}) Q_{\mathcal{O}}^{\dagger}(\phi, \boldsymbol{k}_{\text{d-2}}) = S(R_{\mathcal{O}}(\phi, \boldsymbol{k}_{\text{d-2}})). \tag{B.22}$$

For example, in the presence of time-reversal symmetry Eq. (B.22) establishes that $S$ is isomorphic to an (anti)symmetric matrix $S'$ at time-reversal invariant momenta. To see this, we first observe that $Q_{\mathcal{T}} V_{\mathcal{T}}^* = V_{\mathcal{T}} Q_{\mathcal{T}}^* = \mathcal{T}^2 = \pm 1$ from Eq. (B.20). Then, we use $Q_{\mathcal{T}}^{\dagger} = V_{\mathcal{T}}^* = \pm 1$ in Eq. (B.22) and find

$$\left( S(\phi, \boldsymbol{k}_{\text{d-2}}) V_{\mathcal{T}}^T(\phi, \boldsymbol{k}_{\text{d-2}}) \right)^T = \pm S(-\phi, -\boldsymbol{k}_{\text{d-2}}) V_{\mathcal{T}}^T(\phi, \boldsymbol{k}_{\text{d-2}}). \tag{B.23}$$

At time-reversal invariant momenta, we obtain $S' = \pm S'^T$ for $S' = S V_{\mathcal{T}}^T$.

## C  Factoring out commuting symmetries

Whenever a symmetry group has a unitary symmetry $C_n$ that commutes with all other elements and satisfies $C_n^n(\phi) = C_n^n(0) \exp(i\phi)$, we use the subspaces of $C_n$ to block-diagonalize the reflection matrix. Because the eigenvalues of $C_n$ are proportional to $\exp(i(\phi + 2\pi j)/n)$, for $j \in [0, n)$ the index of the eigenvalue, the periodicity of each of the blocks of the reflection matrix becomes $2\pi n$. A phase change of $2\pi$ corresponds to a map from the $j$th block to the next—the $j+1$th block. Therefore, instead of considering the entire reflection matrix over the $2\pi$ range, we work with only one block over the $2\pi n$ range. We call this procedure factoring out the symmetry $C_n$, because $C_n$ is no longer a symmetry of a single block.

For example, in Sec. 4.1, in addition to the constraints of $C_4$ and $\mathcal{C}$, $C_2 = (C_4)^2$ further constrains the reflection matrix as:

$$V_{C_2}(\phi) r(\phi) Q_{C_2}^{\dagger} = r(\phi), \tag{C.1}$$

where $V_{C_2} = (V_{C_4})^2$ and $Q_{C_2} = (Q_{C_4})^2$. We use this commutation relation to block-diagonalize $r$ into two blocks associated with the $\lambda_{\pm}(\phi) = \pm i \exp(-i\phi/2)$ eigenvalues of $C_2$:

$$U_{V_{C_2}}(\phi) \begin{pmatrix} \lambda_+(\phi) \\ & \lambda_-(\phi) \end{pmatrix} U_{V_{C_2}}^{\dagger}(\phi) r(\phi) U_{Q_{C_2}}(\phi) \begin{pmatrix} \lambda_+(\phi) \\ & \lambda_-(\phi) \end{pmatrix} U_{Q_{C_2}}^{\dagger}(\phi) = r(\phi), \tag{C.2}$$

where $U_{V_{C_2}}(\phi)$ and $U_{Q_{C_2}}(\phi)$ are the eigenvectors of $V_{C_2}(\phi)$ and $Q_{C_2}(\phi)$. Because $[C_2, C_4] = [C_2, \mathcal{C}] = 0$, each block remains constrained by $C_4$ and $\mathcal{C}$. Additionally, because the eigenvalues of $C_2$ swap when $\phi$ changes by $2\pi$, the blocks of $r$ swap as well. Therefore, each block contains the full information of the reflection matrix in the $4\pi$ range. Without loss of generality, we redefine $2\phi \to \phi$ and redefine $r(\phi)$ as one of the blocks:

$$r(\phi) := U_{V_{C_2}, -}^{\dagger}(\phi) r(\phi) U_{Q_{C_2}, -}(\phi). \tag{C.3}$$

The projection in Eq. (C.3) requires applying two different unitaries from the left and right, because $r$ is a linear map from the space of outgoing wavefunctions to the space of incoming wavefunctions.

In Sec. 4.2 we use the same strategy to block-diagonalize the reflection matrix $r$ into two blocks associated with the $\pm i \exp(-i\phi/2)$ eigenvalues of $C_2$. In Sec. 4.3 we block-diagonalize $r$ into two blocks associated with the $\pm i \exp(-i\phi/2)$ eigenvalues of $\mathcal{I}$ at the $k_z = 0, \pi$ lines, which are invariant under inversion.

# D  Implementation of a smooth gauge choice for *V* and *Q*

The expressions for the topological invariants in Eq. (14), (19),  (22) involve integrals of the reflection matrices $r(\phi)$ and $r(\phi, \mathbf{k}_{\text{d-2}})$ over a line. The reflection matrices are, however, linear maps and not operators, making their eigenvalues gauge-dependent and, in general, discontinuous functions of $\phi$ and $\mathbf{k}_{\text{d-2}}$. To compute the invariants we need to choose a gauge for the reflection matrices that makes their eigenvalues continuous functions of $\phi$ and $\mathbf{k}_{\text{d-2}}$.

Because the reflection matrices in the invariant expressions are a result of applying a parameter-dependent basis transformation to the original reflection matrices (see Eq. (C.3)), we need to construct a smooth gauge for the eigenvectors of $V(\phi)$ and $Q(\phi)$. To do this, we first compute $V(\phi)$ and $Q(\phi)$ for a set of $N$ values $\phi \in [0, 2\pi]$, such that they are $2\pi$-periodic. Then, we compute the eigendecomposition of $V(\phi)$ and $Q(\phi)$ for each $\phi$ and separate them into two sets:

$$
\begin{aligned}
V(\phi) &= \lambda_+(\phi) U_{V,+}(\phi) U_{V,+}^\dagger(\phi) + \lambda_-(\phi) U_{V,-}(\phi) U_{V,-}^\dagger(\phi), \\
Q(\phi) &= \lambda_+(\phi) U_{Q,+}(\phi) U_{Q,+}^\dagger(\phi) + \lambda_-(\phi) U_{Q,-}(\phi) U_{Q,-}^\dagger(\phi),
\end{aligned}
\tag{D.1}
$$

where $\lambda_\pm(\phi)$ are eigenvalues of $V(\phi)$ and $Q(\phi)$ smoothly varying with $\phi$, and $U_{V,\pm}(\phi)$ and $U_{Q,\pm}(\phi)$ are the corresponding eigenvectors arranged as columns. The latter are not necessarily smooth functions of $\phi$, and our goal is to construct a smooth gauge for them.

We construct a smooth gauge for each of the four sets of eigenvectors $U_{V,\pm}(\phi)$ and $U_{Q,\pm}(\phi)$ by iterating over $\phi$ and following the steps below:

1. Compute the overlap matrix $O(\phi) = U_{V,+}^\dagger(\phi) U_{V,+}(\phi + \delta\phi)$.

2. Compute the singular value decomposition $O(\phi) = U(\phi) S(\phi) V^\dagger(\phi)$.

3. Compute the gauge transformation matrix $G(\phi) = U(\phi) V(\phi)$.

4. Update the eigenvectors $U_{V,\pm}(\phi + \delta\phi) \rightarrow U_{V,\pm}(\phi + \delta\phi) G^\dagger(\phi)$.

5. Repeat for the next value of $\phi$, until $\phi = 2\pi$.

6. Compute the overlap matrix $O(2\pi)$ and spread the gauge transformation uniformly over all the eigenvectors.

The last step ensures that the gauge is also $2\pi$-periodic. Additionally, if a symmetry that maps $\phi$ to $-\phi$ is present, e.g. time-reversal, we further constrain the gauge by choosing symmetric eigenvectors at $\phi = 0, \pi$. By the end of this procedure, the overlap matrix $O(\phi)$ is the identity matrix in the limit $\delta\phi \rightarrow 0$, making the eigenvectors smooth functions of $\phi$. We repeat the same procedure for the eigenvectors $U_{V,-}(\phi)$ and $U_{Q,\pm}(\phi)$. The code for constructing the smooth gauge is available in Ref. [54].

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
