# Peer review of "Scattering theory of higher order topological phases"

_SciPost Physics, doi:SciPost Phys. 19, 058 (2025)_

## Round 1 · Referee Report · Anonymous (Referee 1) · 2025-3-17

Report

In this manuscript, the authors study the scattering theory for higher-order topological phases and propose a scattering formalism to characterize the intrinsic higher-order topology. They first demonstrate the limitation of previously established scattering invariants and develop the intrinsic scattering invariants to correctly classify the intrinsic higher-order topological phases. Specifically, based on a scattering geometry compatible with the spatial symmetry, they introduce a reflection matrix with a flux to define the topological invariant. Then, they apply their approach to several examples of intrinsic higher-order topological phases even in the presence of disorder.

This work extends the scattering classification of topological phases to intrinsic higher-order band topology protected by spatial symmetry. The results are interesting and the derivation looks sound. I think this work will advance our understanding on the scattering theory of crystalline topology. Therefore, I would like to recommend the publication in SciPost after the authors address the following comments.

1 In Eq. (13), the effective Hamiltonian is obtained from the matrix h(\phi). However, there is no explicit definition of h(\phi) here. It seems to me that h(\phi) is a projection of r(\phi) onto the subspace of symmetry according to subsequent statement. I suggest the authors add some detailed description about the meaning of h(\phi).

2 For the BBH model (Fig. 5), the authors introduce disorder into the intra-cell hopping \lambda and show the universality of the scattering invariant for different geometry and sizes. I have a question that what is the meaning of the value of \lambda in Fig. 5 after adding the randomness into the parameter. I suggest the authors give more details about the form and strength of disorder here.

Recommendation

Publish (easily meets expectations and criteria for this Journal; among top 50%)

---

## Round 1 · Referee Report · Anonymous (Referee 2) · 2025-3-18

Report

In this work, the authors propose a method to generalize the scattering matrix topological invariants of topological insulators to higher order ones. This requires setting up a special transport geometry that preserves all symmetries and maps the bulk Hamiltonian to an effective one related to the reflection matrix. I found the discussion very instructive: the authors first show the standard method fails for extrinsic HOTIs and argue how the transport setup needs to be modified to address intrinsic HOTIs (where there is a bulk invariant and bulk boundary correspondence). Then the authors go over different models for symmetry protected HOTIs and check that the computed invariant matches the known topological phase transition in the bulk. The authors also consider network models where only scattering matrices are defined, and show their method applies in this case too even in the absence of a Hamiltonian.

The work is thorough, generally pedagogical, and describes in detail the context of the field. The results are interesting and robust, and I think it will be a useful addition to the literature. In my opinion the work satisfies criterion number 4 (a breakthrough on a long standing research stumbling block) to be publishable in SciPost Physics, as there was no scattering invariant for intrinsic HOTIs before.

The work is also very formal. If at all possible, it would be interesting to connect these results to transport experiments that can be measured. A flux can be threaded through a cylinder, but twists in boundary conditions are a convenient mathematical tool that does not relate to experiments. While I still think the work should be published either way, I would encourage the authors to comment on how these transport invariants could be measured, or what implications their work has for experiments in general. Measuring these transport invariants is for example possible in topological insulator cylinders under flux, but I assume in this case the analog experiment is not realistic.

Among more technical comments, an aspect of this work that I found confusing on the first two reads is that for C_n symmetric samples, n leads are attached outside and n inside, but each group of n is considered as one lead. This was initially confusing when comparing the figures, like Fig.1 and Fig.3, and I only cleared my confusion by reading the caption of Fig. 3 which explicitly says “all of which are considered as one lead”. Throughout the text, when the plural “leads” is used, it was unclear whether this referred to inner and outer leads, or to the set of leads that is considered one lead. Statements like “the leads are in between the zero modes” can be confusing. It would help to use “sub-leads” when they are should be considered as one lead, for example. I would suggest explaining this in the main text. Essentially, the authors use a version of the Corbino geometry widely used in the Quantum Hall effect, where inner and outer leads are annular (i.e. the touch the whole boundary). Here, the sample has discrete rotation symmetry, so one can use a discrete set of leads, even thought they still play the role of one lead. Explaining this will make every thing clearer. Also, is there a practical constraint from KWANT that does not allow to use a full annular lead?

  • Another confusing statement on pg. 6 is that they apply twisted boundary conditions in the direction perpendicular to the cylinder axis. Shouldn’t this be parallel? The twist mimicks momentum in the direction perpendicular to transport, which occurs from inside to outside. Later on the momentum so obtained is called k_z, so I think this should read “parallel”.

Recommendation

Publish (meets expectations and criteria for this Journal)

---

## Round 2 · Author Response

We thank the referees for their positive evaluations and useful suggestions. We have addressed all their comments in the updated version of the manuscript. We have also made several minor language corrections. We provide the manuscript with the changes highlighted at the URL: https://surfdrive.surf.nl/files/index.php/s/S13PbbRAo3RPLmh.

With the changes implemented, we believe that the manuscript is now suitable for publication in SciPost Physics.

Below is the detailed response to the referees' comments.

Referee 1

1 In Eq. (13), the effective Hamiltonian is obtained from the matrix h(\phi). However, there is no explicit definition of h(\phi) here. It seems to me that h(\phi) is a projection of r(\phi) onto the subspace of symmetry according to subsequent statement. I suggest the authors add some detailed description about the meaning of h(\phi).

The referee is correct in their interpretation, and we now define h(\phi) in the updated version of the manuscript.

2 For the BBH model (Fig. 5), the authors introduce disorder into the intra-cell hopping \lambda and show the universality of the scattering invariant for different geometry and sizes. I have a question that what is the meaning of the value of \lambda in Fig. 5 after adding the randomness into the parameter. I suggest the authors give more details about the form and strength of disorder here.

We have added the explicit definition of disorder to the manuscript. Specifically, we choose the intra-cell hopping equal to $\lambda[1 + \delta(x, y)]$, with $\delta$ a random variable. In other words, $\lambda$ is the average value of the intra-cell hopping.

Referee 2

The work is also very formal. If at all possible, it would be interesting to connect these results to transport experiments that can be measured. A flux can be threaded through a cylinder, but twists in boundary conditions are a convenient mathematical tool that does not relate to experiments. While I still think the work should be published either way, I would encourage the authors to comment on how these transport invariants could be measured, or what implications their work has for experiments in general. Measuring these transport invariants is for example possible in topological insulator cylinders under flux, but I assume in this case the analog experiment is not realistic.

We agree with the referee's observation. While we believe that the main impact of our work is theoretical—namely characterizing disordered HOTIs—there likely exist quantized transport phenomena that relate to the interplay between adiabatic pumping and symmetry eigenvalues. We have added a comment on this in the conclusion of the manuscript.

Among more technical comments, an aspect of this work that I found confusing on the first two reads is that for C_n symmetric samples, n leads are attached outside and n inside, but each group of n is considered as one lead. This was initially confusing when comparing the figures, like Fig.1 and Fig.3, and I only cleared my confusion by reading the caption of Fig. 3 which explicitly says “all of which are considered as one lead”. Throughout the text, when the plural “leads” is used, it was unclear whether this referred to inner and outer leads, or to the set of leads that is considered one lead. Statements like “the leads are in between the zero modes” can be confusing. It would help to use “sub-leads” when they are should be considered as one lead, for example. I would suggest explaining this in the main text. Essentially, the authors use a version of the Corbino geometry widely used in the Quantum Hall effect, where inner and outer leads are annular (i.e. the touch the whole boundary). Here, the sample has discrete rotation symmetry, so one can use a discrete set of leads, even thought they still play the role of one lead. Explaining this will make every thing clearer.

We thank the referee for this observation. We updated the manuscript so that we only refer to the single inner or outer lead and specify that it consists of multiple disjoint regions. We preferred to avoid introducing the term "sub-leads" to keep the terminology simple, but at the same time we avoid referring to the single parts of the inner or outer lead in the updated manuscript.

Also, is there a practical constraint from KWANT that does not allow to use a full annular lead?

Attaching ideal leads that we use in the manuscript to the entire perimeter is directly possible in Kwant. We chose to attach leads to single sites in the manuscript to demonstrate that the intrinsic invariant works even in that case. On the other hand, solving the scattering problem off a flux defect in an infinite 2D system would require extending numerical scattering theory beyond the current state of the art.

Another confusing statement on pg. 6 is that they apply twisted boundary conditions in the direction perpendicular to the cylinder axis. Shouldn’t this be parallel? The twist mimicks momentum in the direction perpendicular to transport, which occurs from inside to outside. Later on the momentum so obtained is called k_z, so I think this should read “parallel”.

We thank the referee for pointing this out. We have corrected the text accordingly.

---

## Round 2 · List of Changes

The full changes are visible at: https://surfdrive.surf.nl/files/index.php/s/S13PbbRAo3RPLmh

---

## Editorial Decision

published